# Structural optimization of siRNA conjugates for albumin binding achieves effective MCL1-directed cancer therapy

Ella N. Hoogenboezem[1], Shrusti S. Patel[1], Justin H. Lo[1,2], Ashley B. Cavnar[2], Lauren M. Babb [1], Nora Francini[1], Eva F. Gbur [1], Prarthana Patil[1], Juan M. Colazo [1,3], Danielle L. Michell [2], Violeta M. Sanchez[2], Joshua T. McCune [1], Jinqi Ma[1], Carlisle R. DeJulius[1], Linus H. Lee[1], Jonah C. Rosch[4], Ryan M. Allen[2], Larry D. Stokes[1], Jordan L. Hill[1], Kasey C. Vickers[2,5], Rebecca S. Cook [1] & Craig L. Duvall [1] ✉

The high potential of siRNAs to silence oncogenic drivers remains largely untapped due to the challenges of tumor cell delivery. Here, divalent lipid-conjugated siRNAs are optimized for in situ binding to albumin to improve pharmacokinetics and tumor delivery. Systematic variation of the siRNA conjugate structure reveals that the location of the linker branching site dictates tendency toward albumin association versus self-assembly, while the lipid hydrophobicity and reversibility of albumin binding also contribute to siRNA intracellular delivery. The lead structure increases tumor siRNA accumulation 12-fold in orthotopic triple negative breast cancer (TNBC) tumors over the parent siRNA. This structure achieves approximately 80% silencing of the anti-apoptotic oncogene *MCL1* and yields better survival outcomes in three TNBC models than an MCL-1 small molecule inhibitor. These studies provide new structure-function insights on siRNA-lipid conjugate structures that are intravenously injected, associate in situ with serum albumin, and improve pharmacokinetics and tumor treatment efficacy.

Short interfering RNA (siRNA) therapies are powerful tools to silence disease-driving genes that are classically "undruggable". However, in their unmodified form, siRNAs undergo rapid renal clearance and nuclease degradation, resulting in poor in vivo target tissue bioavailability. Initial clinical success of siRNA was achieved by the FDA-approved, first-in-class siRNA-encapsulating lipid nanoparticle (LNP) medicine patisiran. This LNP system was optimized to prevent siRNA degradation and mediate hepatic delivery for treatment of hereditary transthyretin-mediated amyloidosis[1]. However, nanoparticle formulations are relatively complex to synthesize and are associated with injection reactions[2,3], necessitating premedication with corticosteroids

and antihistamine, as in the case of patisiran[4]. The challenges of LNP carriers for siRNA delivery might be avoided with carrier-free, chemically modified siRNAs. This notion has been explored successfully, yielding four FDA-approved drugs employing trivalent N-acetylgalactosamine (GalNAc)-based targeting of siRNAs to hepatocytes expressing the asialoglycoprotein receptor (givosiran, lumasiran, inclisiran, vutrisiran). However, clinical approval of siRNAs against extrahepatic targets remains elusive.

The molecular precision provided by siRNA makes it an attractive class of drugs for many diseases including cancers, but oncological siRNA applications have not shown success in clinical trials. This may

[1]Department of Biomedical Engineering, Vanderbilt University, Nashville, TN, USA. [2]Department of Medicine, Vanderbilt University Medical Center, Nashville, TN, USA. [3]Medical Scientist Training Program, Vanderbilt University School of Medicine, Nashville, TN, USA. [4]Department of Chemical and Biomolecular Engineering, Vanderbilt University, Nashville, TN, USA. [5]Department of Molecular Physiology and Biophysics, Vanderbilt University, Nashville, TN, USA. ✉e-mail: craig.duvall@vanderbilt.edu

be due, at least in part, to reliance on the idea that the leaky, aberrant tumor vasculature and insufficient lymphatic drainage of tumors, collectively termed the enhanced permeability and retention (EPR) effect, would enable siRNA nanoparticle delivery to tumors, while avoiding healthy tissues. However, nanoparticles designed to leverage the EPR effect have yielded inconsistent results, particularly upon translation into clinical trials[5–8]. Importantly, siRNA delivered by bulky nanocarriers may have limited efficiency in reaching and permeating tumors that are poorly perfused or do not have large-scale lymphovascular disruptions. In contrast, smaller siRNA conjugates, with structurally-optimized end modifications that improve systemic and cell-level pharmacokinetics, may achieve more homogeneous delivery into tumors[9]. End modifications can be used to promote association with various plasma components as a means to modify siRNA pharmacokinetics and biodistribution[10–12]. Binding to albumin is a promising approach for pharmacokinetic modification because it is abundant in plasma and has a long half-life. Albumin also naturally binds to fatty acids (FAs)[9], which can be readily integrated into siRNA structures, enabling development of siRNAs that are intravenously injected carrier-free and can dock onto serum albumin as an endogenous carrier.

Albumin also has several traits that motivate its use as an endogenous carrier specifically for tumor applications. Albumin accumulates in the tumor interstitium in numerous solid cancer models[13–17]. Albumin exhibits >4-fold higher tumor tissue penetration over nanocarriers[18] and is actively taken up by rapidly dividing tumor cells to meet their high metabolic demand[19–21]. The utility of albumin as a cancer therapeutic carrier is supported by the clinical success of the albumin-based nanoparticle Abraxane for paclitaxel delivery[22]. Moreover, albumin is rescued from renal clearance, an obstacle for systemically administered oligonucleotides[9]. Association with albumin may therefore improve tumor bioavailability of siRNA.

Here we took on the unique goal of constructing a library of siRNA-lipid conjugates in an effort to identify lipid siRNA conjugate structural features that promote preferential in situ albumin association and functional tumor delivery in vivo. The library systematically varies lipid valency, branch point positioning (in divalent lipid-siRNA conjugates), length and phosphorothioate content of the linker, degradability of the linker, and the structure of the albumin-binding lipid. The optimized construct was used to deliver therapeutic siRNA sequences to models of triple negative breast cancer (TNBC), an aggressive breast cancer subtype with a disproportionately high mortality rate and few therapeutic options.

Because TNBCs, a heterogenous group of cancers, lack the traditional molecular targets associated with HER2+ and ER+ breast cancers, molecularly targeted therapeutics are relatively unavailable to patients diagnosed with TNBCs, making chemotherapy the primary treatment course. Approximately 70% of TNBCs treated with neoadjuvant chemotherapy (NAC) harbor residual disease (RD), rapid recurrence and high mortality[23]. Molecular and genomic analyses of TNBCs following NAC revealed *MCL1* among the most commonly altered genes in RD. *MCL1*, an anti-apoptotic oncogene of the BCL2 family is a known driver of treatment resistance in many cancers, including TNBC[24]. Efforts at targeting MCL-1 are being actively pursued, with multiple small molecules entering clinical trials[25]. Therefore, we sought to demonstrate therapeutic proof of concept for the lead siRNA-lipid conjugate structure by applying it to target *MCL1* in TNBC models in vivo.

## Results and discussion
### Divalent lipid modifier improves bioavailability of chemically stabilized siRNAs
We synthesized siRNAs stabilized with alternating 2'F and 2'OMe modifications in a "zipper" pattern and terminal phosphorothioate (PS) linkages (Fig. 1a and Supplementary Fig. 1) to confer endonuclease

and exonuclease resistance[26]. Delivery of zipper- and PS-modified siRNA sequences against luciferase (si$_{Luc}$) inhibited luciferase expression in MDA-MB-231 cells with similar silencing activity to commercially available Dicer substrate siRNAs that sparingly incorporate 2' modifications (Supplementary Fig. 2). However, agarose gel electrophoresis of siRNA following incubation in serum revealed that the zipper- and PS-modified siRNA remained intact, while the Dicer substrate siRNA was mostly degraded within a few hours (Supplementary Fig. 2). Henceforth, all siRNAs harbor the zipper and PS modifications, except studies that focus specifically on variation of PS content.

The structure of hydrophobic modifications on siRNA can be tuned to direct siRNA binding to different serum components, such as lipoproteins and albumin, after intravenous (i.v.) administration, which can consequently modify pharmacokinetics and biodistribution[11,27]. Although there is evidence for active uptake of both albumin[13–17] and lipoproteins[28–30] by tumor cells, we sought to screen for siRNA structures that preferentially bind to and leverage the carrier properties of endogenous albumin because of its smaller size, lower relative liver tropism, and its capacity for re-absorption back into circulation in the kidney, evading excretion via urine[11]. Human albumin has a high in vivo plasma concentration (35–55 mg/ml) and long half-life (~19 days), making it an ideal candidate for targeted binding in plasma to serve as an endogenous carrier that increases therapeutic bioavailability[9].

A library of siRNA-lipid conjugates was generated using solid phase synthesis to maximize yield, purity, and reproducibility, while enabling controlled variation of the linker structure. Reports suggest that valency of lipid end-modified siRNA conjugates affects bioavailability and pharmacodynamics[31–33]. Thus, we compared pharmacokinetics of i.v.-delivered siRNAs conjugated to monovalent (L$_1$) or divalent (L$_2$) lipid tails comprising 18-carbon stearyls; stearyl was chosen due to its higher affinity to albumin than shorter lipids[34–36]. Absolute circulation half-life ($t_{1/2}$) of Cy5-labeled siRNA-L$_2$, measured in real time using intravital fluorescence vascular imaging, was $46 \pm 5.9$ min; the divalent structure showed higher $t_{1/2}$ than siRNA-L$_1$ ($28 \pm 4.2$ min) despite no difference in percent association with albumin in plasma harvested from mice ~1 h after i.v. injection (Supplementary Fig. 3). Whole organ fluorescence was used to assess siRNA-L$_1$ and siRNA-L$_2$ biodistribution 45 min following i.v. delivery, showing decreased renal loss of siRNA-L$_2$ as compared to siRNA-L$_1$. Due to its longer circulation time and lower kidney biodistribution, the zipper- and PS-stabilized divalent siRNA-L$_2$ construct was used as the basis for additional structure-function exploration.

### Hydrophilic linker length increases albumin affinity while reducing self-assembly of lipid-siRNA conjugates
Variants of siRNA-L$_2$ were synthesized with 0 to 30 ethylene glycol (EG) repeats [siRNA < (EG$_X$L)$_2$] within the hydrophilic linker bridging the lipid and siRNA moieties; structures were created in increments of 6 using a hexaethylene phosphoramidite (Fig. 1a, b). For comparative reference, two previously reported siRNA-lipid conjugates were included: cholesterol-TEG-siRNA[37–39] (si-EG$_3$-chol) and si-PEG$_{45}$ < L$_2$[40]. Electrophoretic mobility shift assay (EMSA) assessing albumin association revealed the shifted mobility of si < (EG$_0$L)$_2$, si < (EG$_6$L)$_2$, si < (EG$_{18}$L)$_2$, and si < (EG$_{30}$L)$_2$ following incubation with human albumin, while mobility of free siRNA was unaltered by albumin (Fig. 1c), confirming that all siRNA < (EG$_X$L)$_2$ conjugates have some level of association with albumin. Similar results were observed using mouse serum albumin (Supplementary Fig. 4). The si-EG$_3$-chol benchmark compound also displayed albumin-dependent mobility shifts (Fig. 1c). However, mobility shifting of si-PEG$_{45}$ < L$_2$ was seen in both the presence and absence of albumin, suggesting that, in addition to binding to albumin, si-PEG$_{45}$ < L$_2$ self-associates into micelles.

Biolayer interferometry (BLI) was used to measure albumin association and dissociation kinetics of the si < (EG$_X$L)$_2$ variants (Fig. 1d and Supplementary Fig. 4). As expected, free siRNA did not exhibit albumin

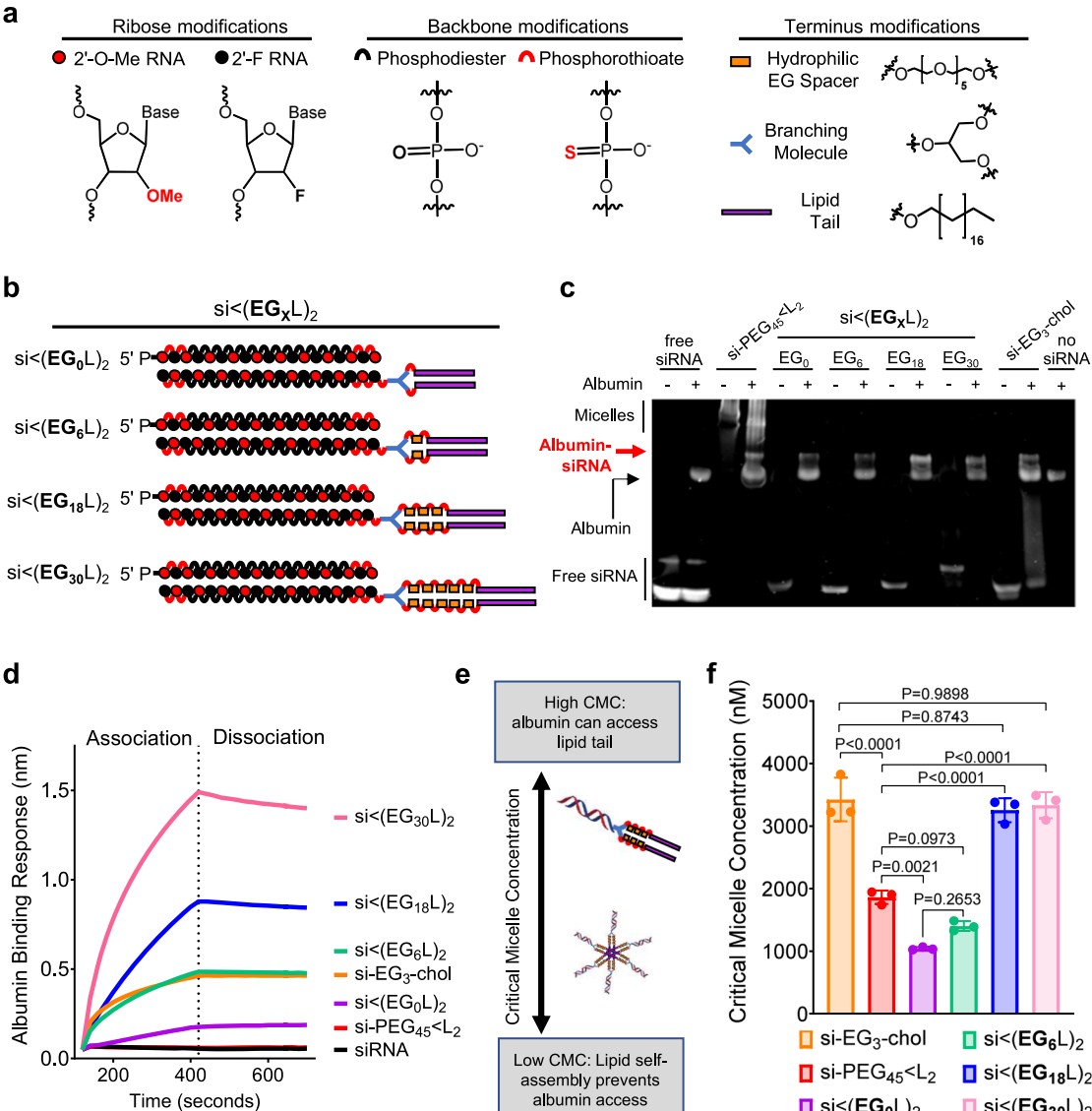

**Fig. 1 | Increased linker EG repeats increases si < (EGₓL)₂ albumin binding by reducing self-association.** Schematic structural representations of the siRNA and lipid components (**a**) and linker EG repeats (**b**) comprising the novel siRNA-lipid conjugates si < (EGₓL)₂. **c** Representative image of EMSA analysis, used to assess si < (EGₓL)₂ interaction with human albumin. **d** Representative in vitro binding responses of siRNA conjugates to human serum albumin (400 nM) measured by biolayer interferometry. **e** Model illustration of how micellization of siRNA-lipid conjugates, dictated by critical micelle concentration (CMC), influences availability of the lipids for albumin binding. **f** Critical micelle concentration of siRNA conjugates observed after 2 h of incubation with Nile Red at 37 °C. Each bar represents the average CMC (± S.D.), and each point represents the value measured for an individual experimental replicate, n = 3. Significance was determined by one-way ANOVA with Tukey's multiple comparisons test.

binding. Increasing EG linker length progressively increased the affinity of si < (EGₓL)₂ for albumin, with both si < (EG₁₈L)₂ (KD = 30 ± 0.3 nM) and si < (EG₃₀L)₂ (KD = 9.49 ± 0.1 nM) surpassing what was seen with si-EG₃-chol. These findings are consistent with previous reports showing the influence of linker architecture on the albumin affinity of semaglutide, a therapeutic GLP-1 agonist[41], and highlight the impact of the EG repeats within the linker region.

Amphiphilic lipid-modified nucleic acids may self-assemble into micellar structures[42–44], potentially interfering with albumin association if the albumin-binding lipid tails sequester in the micellar core (Fig. 1e). To establish the impact of linker length on si < (EGₓL)₂ self-assembly, the critical micelle concentration (CMC) was determined (Fig. 1f and Supplementary Fig. 5). As predicted by cholesterol's relatively bulky structure that interferes with tight packing, a relatively high CMC was measured for si-EG₃-chol (3430 ± 350 nM)[45], while si-PEG₄₅ < L₂ exhibited a lower CMC (1860 ± 60 nM), indicating that si-

PEG₄₅ < L₂ is more likely to self-associate. Interestingly, we observed a correlation between number of EG repeats and CMC, with si < (EG₀L)₂ having the lowest CMC (1040 ± 23 nM), and si < (EG₁₈L)₂ and si < (EG₃₀L)₂ harboring the highest CMCs (3260 ± 190 nM and 3330 ± 210 nM). These results demonstrate that increasing the EG spacer length diminishes the propensity for siRNA-lipid micellization, an important consideration when employing long FA chains for albumin binding in situ.

## Hydrophilic linker length determines pharmacokinetics and in vivo plasma disposition of lipid-siRNA conjugates

Cy5-labeled si < (EGₓL)₂ variants were administered i.v. to assess linker effects on pharmacokinetics. Intravital fluorescence microscopy of mouse ear vasculature demonstrated the rapid and complete diminution of circulating free siRNA within 30 min, while si-EG₃-chol and si-PEG₄₅ < L₂ retained some observable circulating siRNA at

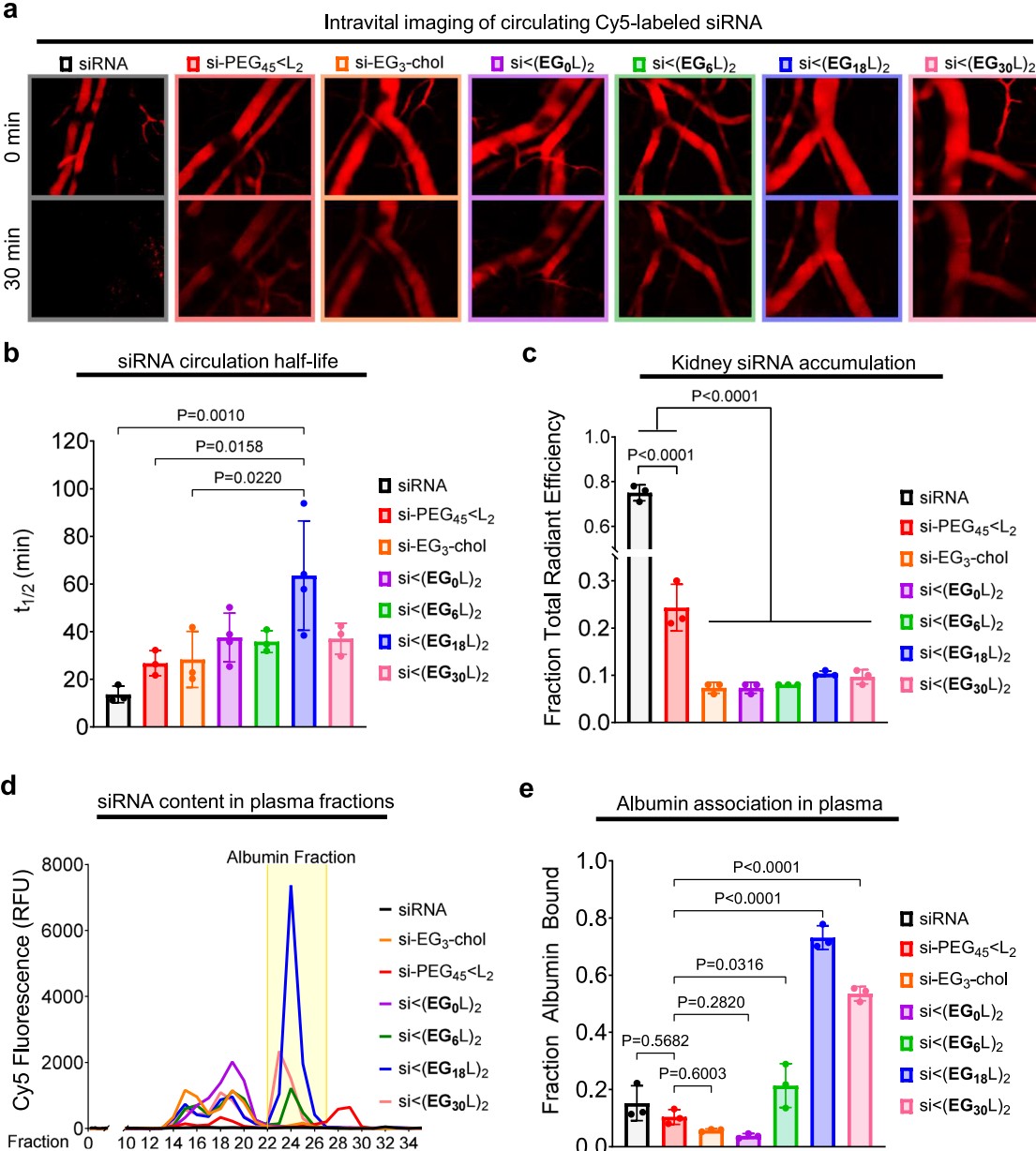

**Fig. 2 | The si < (EG₁₈L)₂ structure achieves the longest circulation half-life and highest albumin association in plasma.** Cy5-labeled conjugate or parental siRNA structures were delivered i.v. to mice at 1 mg/kg. **a** Representative images of intravascular Cy5 fluorescence measured in mouse ear vasculature at 0 min and 30 min after treatment are shown. **b** Intravascular Cy5 fluorescence recorded over ~1 h following delivery was used to calculate absolute siRNA circulation half-life ($t_{1/2}$). Each point represents the half-life calculated from an individual mouse ($n = 3$–$4$). **c** Cy5 fluorescence in kidneys, lungs, liver, spleen, and heart was measured ~1 h after treatment. Fluorescence values in kidneys relative to the total fluorescent values of all organs combined is shown; each point represents the ratio from an individual mouse ($n = 3$). **d, e** Plasma samples collected from mice ~1 h after treatment were fractionated and quantitated for Cy5 fluorescence by size exclusion chromatography. A representative trace is shown of fluorescence signal within each plasma fraction for each siRNA structure; the albumin-containing plasma fractions are highlighted. **d** Cy5 fluorescence within the albumin-containing fractions was quantitated relative to total plasma fluorescence. Each point represents the plasma taken from an individual mouse ($n = 3$) (**e**). For (**b, c, e**), each bar represents the average value ($\pm$ S.D.). For (**b, c**), significance was determined using one-way ANOVA with Tukey's multiple comparisons test. For (**e**), significance was determined by one-way ANOVA with Dunnett's multiple comparisons test using si-PEG₄₅ < L₂ as the comparison group.

30 min, as did each of the si < (EGₓL)₂ variants (Fig. 2a). Real-time measurements of circulating Cy5 fluorescence over time enabled calculation of absolute half-life ($t_{1/2}$), demonstrating increasing $t_{1/2}$ with increasing EG repeats, with the $t_{1/2}$ of si < (EG₁₈L)₂ nearly 5-fold higher than the parent siRNA duplex. Interestingly, the $t_{1/2}$ for si < (EG₃₀L)₂ was less than what was seen for si < (EG₁₈L)₂ (Fig. 2b and Supplementary Table 2), suggesting a limit to the positive impact of increasing linker length on circulation half-life.

Approximately 1 h after injection, siRNA biodistribution was measured as Cy5 fluorescence in whole organs, revealing abundant renal accumulation of free siRNA, but substantially reduced renal accumulation of all si < (EGₓL₂) variants (Fig. 2c and Supplementary Fig. 5). Interestingly, the previously described si-PEG₄₅ < L₂ exhibited more renal accumulation than the new si < (EGₓL₂) variants. This observation may be due to hydrolytically degradable ester bonds within the DSPE-PEG₂₀₀₀ used for si-PEG₄₅ < L₂ synthesis. Ester

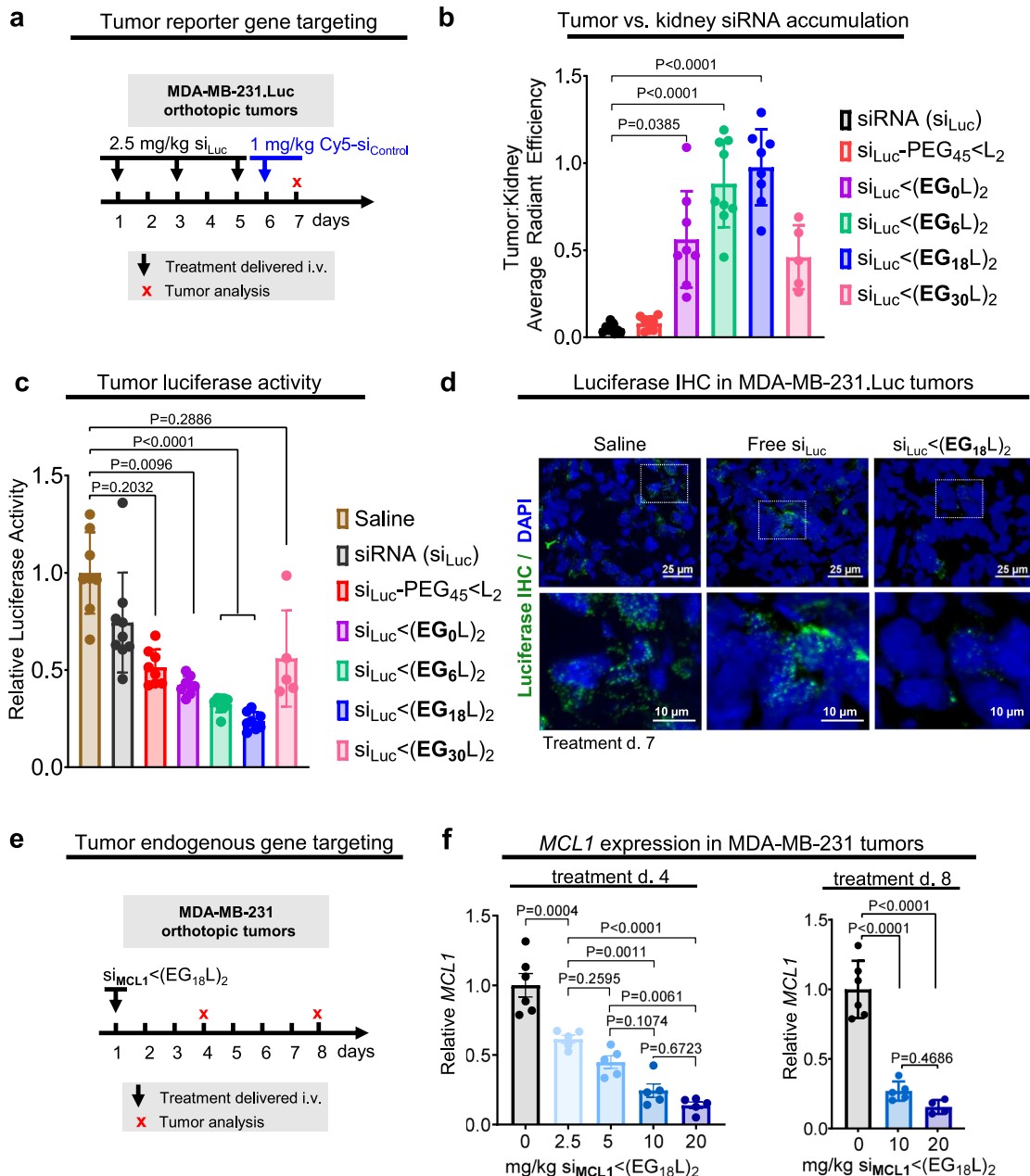

**Fig. 3 | Robust tumor accumulation and tumor gene silencing using si < (EG18L)2. a–d** Mice bearing intra-mammary MDA-MB-231.Luc tumors were treated with siRNA-lipid conjugates, or free siRNA, using a siRNA sequence against luciferase (siLuc). **a** Schematic showing treatment schedule for i.v. delivery of siRNA-lipid conjugates, or free siRNA, to tumor-bearing mice. **b** Cy5 fluorescence tumor:kidney ratio for each mouse as measured on day 7 (18 h after injection with fluorescent siRNA) (*n* = 5–9). **c** Luciferase activity was measured in tumor lysates collected on treatment day 7. Bars show the average (± S.D.) tumor luciferase per group, and points are the values for each tumor (assessed in triplicate) (*n* = 5–9). All values shown are relative to the average luciferase activity of saline-treated controls, which was set to a value of 1. **d** Tumors were assessed by IHC analysis for firefly luciferase (green). Nuclei were counterstained with DAPI (blue). Representative images are shown (**e**, **f**). Mice bearing intra-mammary MDA-MB-231 tumors were treated one time with si < (EG18L)2, using siRNA sequences against *MCL1* (siMCL1) at increasing doses (2.5 mg/kg–20 mg/kg). Tumors were assessed on treatment day 4 and day 8. **e** Schematic showing i.v. delivery of siRNA-lipid conjugates, or free siRNA, to tumor-bearing mice. **f** Human *MCL1* mRNA levels measured in whole tumor RNA collected on treatment day 4 (left panel) and day 8 (right panel). Bars show the average (± S.D.), and points are the values for an individual tumor (assessed in triplicate) (*n* = 5–6). All values shown are relative to values measured with saline controls, which were set to a value of 1. For (**b**, **c**), significance was determined using one-way ANOVA with Dunnetts's multiple comparisons test using the siRNA and saline groups for comparison, respectively. For (**f**), significance was determined using one-way ANOVA with Tukey's multiple comparisons test.

hydrolysis might release siRNA from albumin prematurely in circulation, resulting in renal clearance and shorter circulation time relative to the more stable si < (EGxL2) variants. Albumin itself harbors intrinsic esterase activity, making the hydrolytic stability of albumin-interacting drugs particularly important[46]. However, si < (EGxL2) variants likely remain albumin-bound, thus evading renal clearance. This hypothesis

was tested by size exclusion chromatography of plasma collected from mice after i.v. delivery of the siRNA conjugates. Using the methods exemplified in Supplementary Fig. 6, it was observed that only ~10% of si-PEG45 < L2 associated with the albumin-containing fraction of plasma at 1 h following i.v. treatment, similar to what was detected for free siRNA (Fig. 2d, e). In contrast, increasing the number of EG repeats in

the hydrophilic linker within the si < $(EG_xL_2)$ series increased association with the albumin-containing plasma fractions. Approximately 75% of si < $(EG_{18}L)_2$ co-eluted within the albumin-containing peaks (Fig. 2e), consistent with the high albumin affinity of si < $(EG_{18}L)_2$ measured by biolayer interferometry (Fig. 1d) and in support of the idea that albumin-binding in vivo extends circulation half-life (Fig. 2b). Interestingly, the percentage of si < $(EG_{30}L)_2$ associated with the albumin-containing fractions was decreased compared to si < $(EG_{18}L)_2$. While si < $(EG_{30}L)_2$ shows greater than 50% albumin association in vivo, it also partially elutes with earlier fractions associated with large lipoproteins. Therefore, while both si < $(EG_{18}L)_2$ and si < $(EG_{30}L)_2$ demonstrate high affinity for purified albumin in vitro, si < $(EG_{18}L)_2$ shows a greater selectivity for albumin association in the complex, in vivo environment, suggesting that association with circulating albumin in vivo (Fig. 2d, e) is a better predictor of circulation half-life (Fig. 2b) versus affinity to purified albumin in vitro (Fig. 1d).

### Albumin binding increases tumor accumulation of siRNA-lipid conjugates and tumor gene silencing

To assess tumor accumulation of si < $(EG_XL)_2$ variants, female athymic mice harboring orthotopic Luc-expressing MDA-MB-231 TNBC tumors were treated (2.5 mg/kg, i.v.) on days 1, 3, and 5 with a $si_{Luc}$ sequence synthesized into the different si < $(EG_XL)_2$ variants (Fig. 3a). Mice were treated (1 mg/kg, i.v.) on day 6 with inactive, Cy5-labeled si < $(EG_XL)_2$ for measuring whole tissue fluorescence in tumors and kidneys -18 h after treatment. These studies revealed abundant kidney fluorescence in mice treated with free siRNA or si-$PEG_{45}$ < $L_2$, and little fluorescence in tumors (Fig. 3b). However, progressively increased tumor fluorescence with conversely decreasing kidney fluorescence was observed in mice treated with si < $(EG_0L)_2$, si < $(EG_6L)_2$, and si < $(EG_{18}L)_2$, resulting in a tumor:kidney fluorescence ratio of 0.5, 0.9, and 1.0, respectively. These data suggest that higher association with circulating albumin in vivo may improve both siRNA circulation time and siRNA tumor accumulation, and that these outcomes can be optimized by tuning of the EG linker repeat length. These observations are in line with findings that increasing circulation time enables higher tumor exposure and interaction with target cells[47,48]. Comparing the two variants with longer linkers, tumor to kidney fluorescence was lower in mice treated with si < $(EG_{30}L)_2$ compared to si < $(EG_{18}L)_2$, consistent with its diminished association with the albumin fraction in vivo.

Whole tumor luciferase activity was measured as a readout for siRNA silencing activity in tumors collected 18 h after final treatment (day 7), revealing nearly 80% luciferase activity reduction in the tumors of si < $(EG_{18}L)_2$ treated mice (Fig. 3c). These results were confirmed by immunohistochemical (IHC) staining for luciferase (Fig. 3d and Supplementary Fig. 7). Other reported examples of carrier-free siRNA delivery[47,49–51], including cholesterol-siRNA[52] (10 mg/kg) and receptor-targeted Centyrin-siRNA[47] (3 × 10 mg/kg), produced target gene knockdown in tumors, but none as potent as si < $(EG_{18}L)_2$.

To assess knockdown of the endogenous oncogenic driver *MCL1* in tumors, si < $(EG_{18}L)_2$ was synthesized using *MCL1* siRNA sequences ($si_{MCL1}$). A single bolus was delivered i.v. to MDA-MB-231 tumor-bearing mice (Fig. 3e), revealing dose-dependent *MCL1* knockdown in tumors harvested at either 4 or 8 days after treatment (Fig. 3f), with as much as 85% *MCL1* knockdown as detected by the Quantigene assay. Although the *MCL1* siRNA sequenced used herein was designed to target both human and mouse *MCL1*, we found less than 20% *MCL1* knockdown in livers of mice treated at the highest dose (Supplementary Figs. 8 and 9). Given that the si < $(EG_{18}L)_2$ structure shows the most promise based on a combination of albumin binding, circulation time, tumor accumulation, and tumor gene silencing over other candidates investigated here, our subsequent analyses further interrogate the structural components of si < $(EG_{18}L)_2$.

### Phosphorothioate linkages in the 5' sense and linker region contribute to albumin association and pharmacokinetics

Stabilization of the 5' sense terminus with phosphorothioate (PS) linkages in lieu of phosphodiester (PO) linkages reportedly confers exonuclease resistance, while enabling extrahepatic, carrier-free gene silencing applications[53,54]. A variant of si < $(EG_{18}L)_2$ was synthesized, lacking the PS bond at the 5' sense (Se) terminus (No 5'Se PS). In another variant, PS bonds were also removed from the bonds within the linker (i.e., between the EG units) between the siRNA and the stearyl groups (No 5'Se or Binder PS) (Fig. 4a). PS bond removal from the linker modestly but significantly increased the CMC, suggesting a lower tendency to self-assemble (Supplementary Fig. 10). Although albumin binding affinity was unaffected by PS bond removal in either variant, intravital fluorescence microscopy of circulating Cy5-labeled siRNA variants revealed that removal of PS bonds from both the 5' sense terminus and linker region profoundly diminished $t_{1/2}$ and reduced association with plasma albumin by nearly 10-fold (Fig. 4b). However, removal of the PS from the 5' sense terminus alone had only minimal impact on plasma albumin association. Whole tissue Cy5 fluorescence -1 h post-delivery showed that loss of 5' sense terminus PS bond alone modestly but significantly diminished liver siRNA accumulation compared to parental si < $(EG_{18}L)_2$ (Fig. 4c). Because kidney accumulation was similar for all variants tested, we speculate that the diminished $t_{1/2}$ of variants lacking PS bonds is most likely due to reduced albumin association rather than increased degradation. This conclusion is also supported by prior reports that PS backbones increase oligonucleotide affinity for plasma proteins[55,56]. Due to poor circulation half-life of the si < $(EG_{18}L)_2$ variant lacking PS bonds in both the 5' sense terminus and linker regions, we compared tumor siRNA accumulation and activity of the parental si < $(EG_{18}L)_2$ construct and the variant lacking only the 5' sense PS (Fig. 4d). These studies revealed no significant differences in tumor accumulation and cell uptake, but si < $(EG_{18}L)_2$ with no 5' sense PS bonds exhibited significantly diminished tumor gene silencing (Fig. 4e, f). Whereas our lead construct demonstrated nearly 70% knockdown of the target gene, the elimination of the 5' sense PS bonds resulted in only about 20% knockdown. The comparable biodistribution and uptake of both compounds but large disparity in knockdown suggests that these bonds are stable during circulation and extravasation. However, the PO bonds are likely degraded after biodistribution to the tumor, removing the lipid end groups, and reducing siRNA interactions with and passage through cell and endosomal membranes.

### Proximal positioning of divalent lipid branching point, not relative EG content, dictates tendency to associate with circulating albumin over self-assembly

We next assessed the impact of divalent si < $(EG_{18}L)_2$ conjugate branchpoint position on albumin binding and micellization. It was hypothesized that branchpoint placement between the EG linker repeats and the stearyl groups, rather than between the siRNA and the EG repeats, will constrain the two stearyls and consequently increase self-assembly. We generated two new analogs of si < $(EG_{18}L)_2$ with a distal branchpoint location, one matched for overall EG content (si-$EG_{36}$ < $L_2$) and one matched for the number of EG repeats between the siRNA the lipids (si-$EG_{18}$ < $L_2$) (Fig. 5a). Interestingly, BLI measurement showed that both new branching point variants had lower albumin binding affinity relative to the parent construct si < $(EG_{18}L)_2$ (Fig. 5b). The CMC, measured by Nile Red encapsulation, was significantly reduced for si-$EG_{36}$ < $L_2$ (1838 ± 117 nM) and si-$EG_{18}$ < $L_2$ (2293 ± 132 nM) compared to si < $(EG_{18}L)_2$ (3255 ± 192 nM) (Supplementary Fig. 11 and Fig. 5c), suggesting that the proximal branch site of si < $(EG_{18}L)_2$ increases albumin association due, in part, to a decreased tendency for self-assembly.

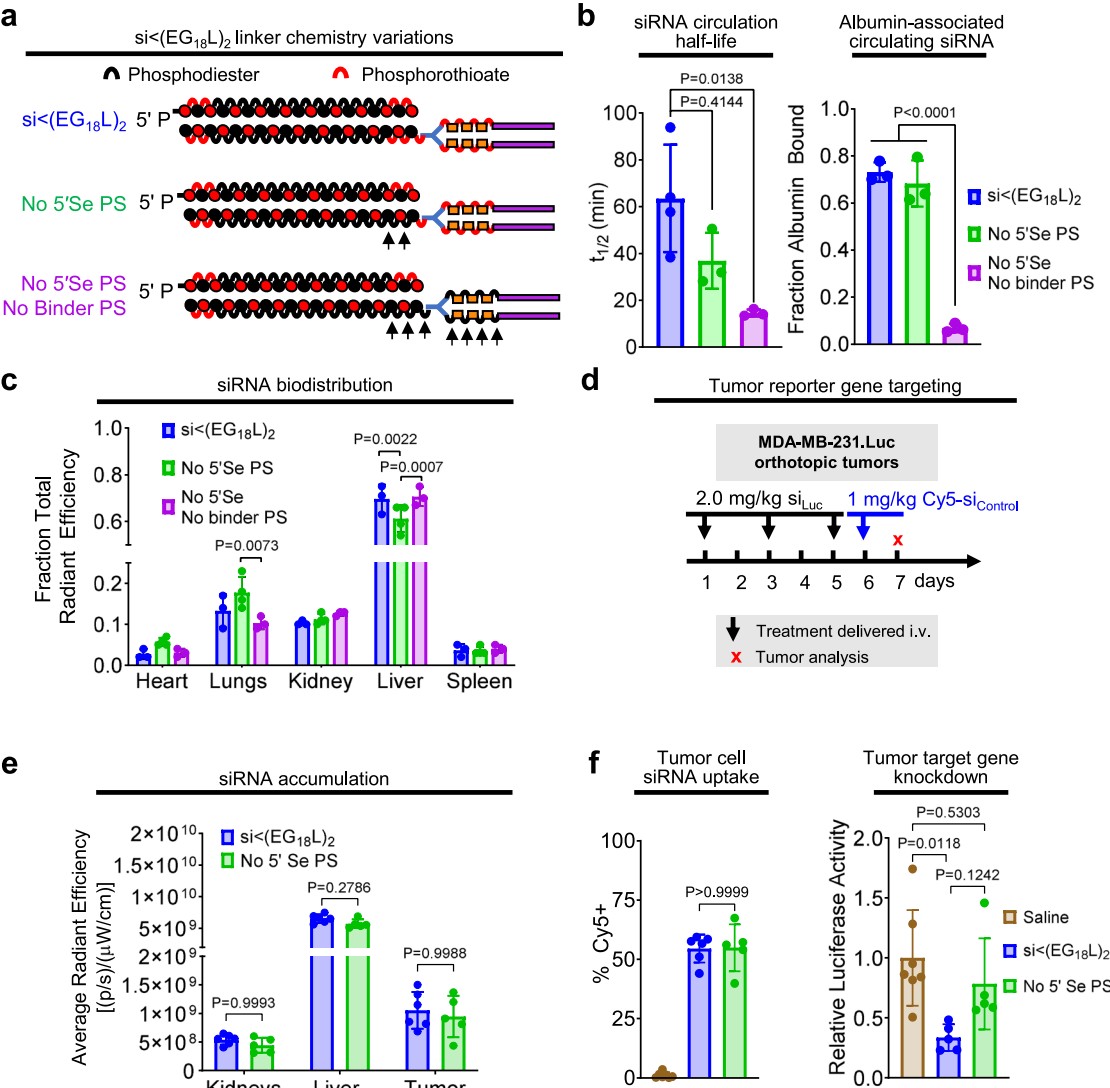

**Fig. 4 | Phosphorothioate linkages on the 5' end of the sense strand and within the linker region contribute to albumin association in plasma and tumor gene silencing activity. a** Schematic structural representations of the siRNA-lipid conjugates with varied inclusion of PS bonds. **b** Cy5-labeled lipid-siRNA conjugates were delivered i.v. to mice at 1 mg/kg. Intravascular Cy5 fluorescence measured through ~1 h following delivery was used to calculate siRNA circulation half-life (left panel) ($n = 3$–4). Size exclusion chromatographic calculation of fraction of Cy5 fluorescence associated with albumin-containing plasma fractions (right panel) ($n = 3$). **c** Cy5 fluorescence in kidneys, lungs, liver, spleen and heart was measured 1 h after treatment. Fluorescence values in each organ relative to the total fluorescent values of all organs combined is shown. **d–f** Mice bearing orthotopic MDA-MB-231.Luc tumors were treated i.v. with siRNA-lipid conjugates, using a siRNA sequence against luciferase (si$_{Luc}$). **d** Schematic showing treatment schedule for i.v. delivery of siRNA-lipid conjugates. **e** Cy5 fluorescence in kidneys, liver, and tumor for each mouse was measured 18 h after treatment ($n = 5$–6). **f** Tumors were collected on day 7 and either dissociated to assess cell-associated Cy5 fluorescence by flow cytometry (left panel) or lysed to assess for luciferase activity (right panel). Bars show the average (± S.D.) value per group, and points are the values for each tumor analyzed. All relative values shown are calculated relative to the average value of saline-treated controls, which was set to a value of 1. Each bar represents the average value (± S.D.), and each point represents values measured for each individual sample ($n = 5$–6). For (**b**, **f**), significance was assigned using one-way ANOVA with Tukey's multiple comparisons test. For (**c**, **e**), significance was assigned by two-way ANOVA with Tukey's multiple comparisons test. For (**e**) and (**f**), $n = 5$–6.

Significantly higher liver accumulation of si-EG$_{18}$<L$_2$ and si-EG$_{36}$<L$_2$ was seen at 1 h post-delivery as compared to si<(EG$_{18}$L)$_2$ (Fig. 5d) an observation that can be attributed to serum lipoprotein binding, as opposed to albumin binding[27]. Although no significant difference in $t_{1/2}$ was seen among the branching architecture variants (Fig. 5e), only about 5% of the delivered si-EG$_{18}$<L$_2$ and si-EG$_{36}$<L$_2$ conjugates was associated with the plasma albumin fraction in vivo, versus ~75% of the parental conjugate (Fig. 5f, g). These data show that the novel, proximal branchpoint placement used in the discovery of si-EG$_{18}$<L$_2$ substantially diminishes self-assembly and enhances albumin association over commonly used distal branch points that are directly adjacent to the multivalent lipids[31,33,34,40].

## Hydrophobicity of lipid and not just high albumin binding affinity is important for conjugate performance

To determine if the nature of the C$_{18}$ lipid affects the performance of si<(EG$_{18}$L)$_2$, we synthesized a variant with a carboxyl on the terminus [si<(EG$_{18}$L$_{diacid}$)$_2$] (Fig. 6a). Although the FA carboxyl functionality is usually consumed in conjugation reactions, screening of various peptide structures that led to the development of GLP-1 agonist drug Semaglutide demonstrated that maintaining a free terminal carboxyl increases albumin affinity and circulation half-life[41]. We also generated a variant with a double bond (si<(EG$_{18}$L$_{unsaturated}$)$_2$), based on reports of higher affinity between albumin and oleate, an unsaturated variant of stearate; testing of oleate lipids was also motivated for the

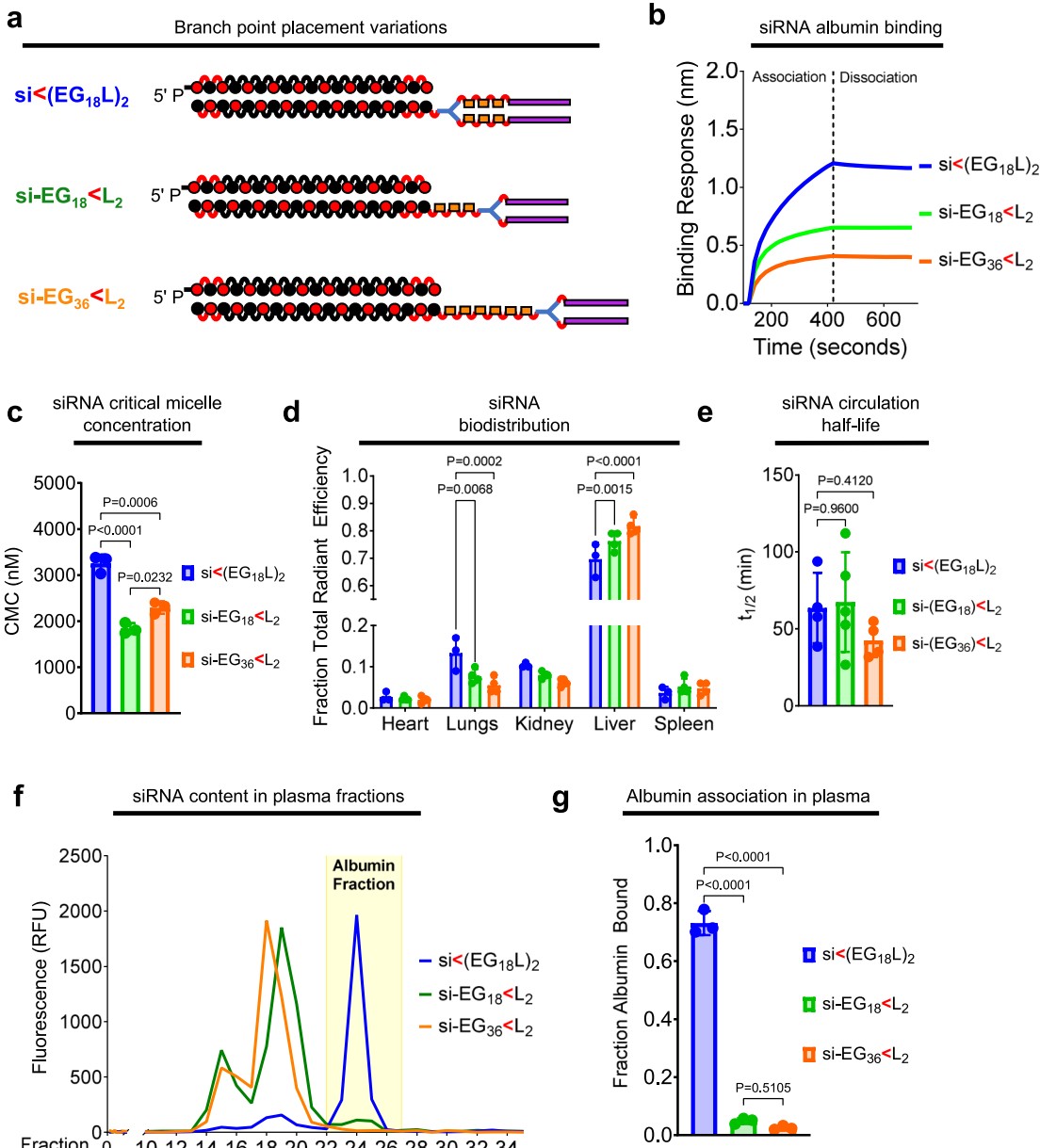

**Fig. 5 | Proximal placement of the divalent lipid branch point immediately adjacent to the siRNA drives albumin binding. a** Schematic structural representations of the siRNA-lipid conjugates with varied branchpoint placement and length of the linker. **b** Representative in vitro binding responses of siRNA conjugates to human serum albumin (400 nM) measured by biolayer interferometry. **c** Critical micelle concentration of siRNA conjugates observed after 2 h of incubation with Nile Red at 37 °C ($n = 3$). **d** Cy5 fluorescence in kidneys, lungs, liver, spleen and heart was measured ~1 h after i.v. delivery of siRNA-lipid conjugates (1 mg/kg). Fluorescence values in each organ relative to the total fluorescent values of all organs combined is shown ($n = 3–4$). **e** Intravascular Cy5 fluorescence measured through 1 h following i.v. delivery of siRNA-lipid conjugate variants (1 mg/kg) was used to calculate siRNA circulation half-life ($t_{1/2}$) ($n = 4–5$). **f** Plasma samples collected from mice ~1 h after treatment (1 mg/kg) were fractionated and quantitated for Cy5 fluorescence by size exclusion chromatography. A representative trace is shown of fluorescence signal within each plasma fraction for each siRNA structure; the albumin-containing plasma fractions are highlighted. **g** Cy5 fluorescence within the albumin-containing fractions was quantitated relative to total plasma fluorescence ($n = 3$). For (**c**–**e**, **g**), each bar represents the average value (± S.D.), and each point represents a value measured for an individual sample. For (**c**, **e**, **g**), significance was assigned using one-way ANOVA with Tukey's multiple comparisons test. For (**d**), significance was assigned using two-way ANOVA with Tukey's multiple comparisons test.

integration of structural "kinks" caused by FA double bonds that may prevent close packing and thus deter self-assembly[57–60]. First, si < (EG$_{18}$-amine)$_2$ was synthesized, to which amine-reactive pentafluorophenyl-modified lipid variants with carboxyl or double bond features were conjugated (Supplementary Fig. 12). Although $t_{1/2}$ and kidney accumulation were similar in si < (EG$_{18}$L$_{diacid}$)$_2$ and si < (EG$_{18}$L$_{unsaturated}$)$_2$ as compared to si < (EG$_{18}$L)$_2$ (Fig. 6b, c), si < (EG$_{18}$L$_{unsaturated}$)$_2$ exhibited diminished association with the albumin-containing plasma fraction in vivo compared to its parental (i.e., saturated) and diacid

counterparts (~40% bound versus ~75–80% bound) (Fig. 6d). Strikingly, si < (EG$_{18}$L$_{diacid}$)$_2$ showed a substantially increased affinity for human albumin (Fig. 6e), nearly two orders of magnitude lower KD than the parental si < (EG$_{18}$L)$_2$ (Supplementary Fig. 13). To determine whether the impact of this increased affinity for albumin would extend the circulation time of si < (EG$_{18}$L$_{diacid}$)$_2$, we measured circulating Cy5 fluorescence through 24 h following delivery at 5 mg/kg siRNA but found no significant difference between circulation time of si < (EG$_{18}$L$_{diacid}$)$_2$ or si < (EG$_{18}$L)$_2$ (Supplementary Fig. 13).

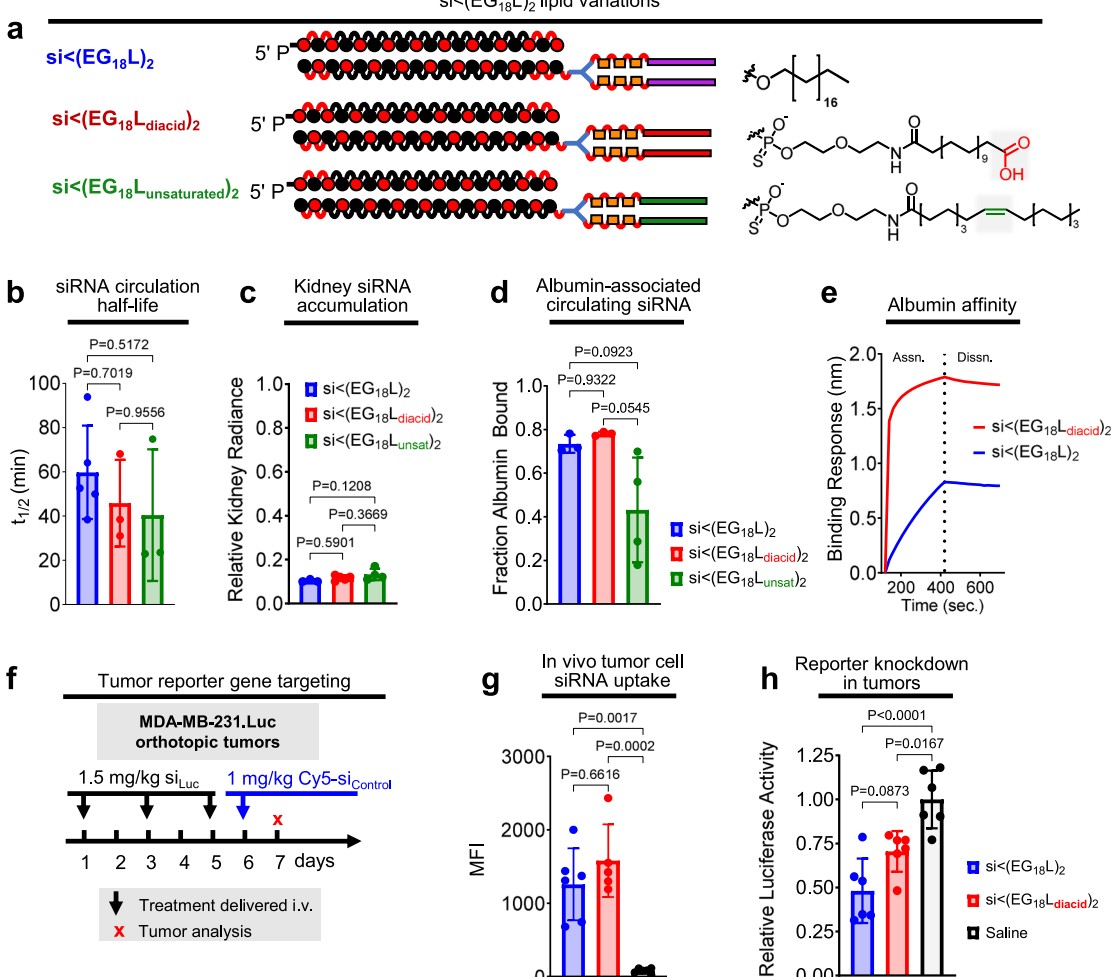

**Fig. 6 | Effects of unsaturated and carboxy-terminated lipid structures.**
**a** Structures of the siRNA-lipid conjugates with varied lipid chemistries. **b** Cy5-labeled siRNA-lipid conjugates were delivered i.v. at 1 mg/kg. Intravascular Cy5 fluorescence was measured over ~1 h and used to calculate siRNA circulation half-life ($n = 3-5$). **c** Cy5 fluorescence in kidneys, lungs, liver, spleen, and heart was measured 1 h after treatment. Fluorescence values in kidneys relative to the total fluorescence values of all organs combined is shown ($n = 3-5$). **d** Size exclusion chromatographic calculation of fraction of Cy5 fluorescence associated with albumin-containing plasma fractions ($n = 3-4$). **e** Representative in vitro binding responses of siRNA conjugates to human serum albumin (400 nM) measured by biolayer interferometry. **f** Schematic showing treatment schedule for i.v. delivery of siRNA-lipid conjugates. Tumors collected on day 7 were dissociated, and tumor cells were assessed for Cy5 fluorescence by flow cytometry ($n = 5-6$) (**g**) or lysed and assessed for luciferase activity ($n = 6$) (**h**). Each bar represents the average value ($\pm$ S.D.), and each point represents values measured for each independent tumor sample. For (**b–c**, **g**, **h**), significance was assigned using one-way ANOVA with Tukey's multiple comparisons test. For **d**, significance was assigned using a Kruskal-Wallis test with Dunn's multiple comparisons test.

We further tested si < (EG$_{18}$L)$_2$ and si < (EG$_{18}$L$_{diacid}$)$_2$ in mice bearing Luc-expressing MDA-MB-231 tumors, using the low dose of 1.5 mg/kg in an attempt to dissect any subtle differences in silencing potency (Fig. 6f). Conjugate si$_{Luc}$ sequences were administered i.v. on days 1, 3, and 5. On day 6, an inactive, Cy5-labeled conjugate was injected for uptake measurement purposes. Interestingly, si < (EG$_{18}$L$_{diacid}$)$_2$ demonstrated higher renal accumulation and lower liver accumulation compared to the si < (EG$_{18}$L)$_2$ (Supplementary Fig. 14). Tumors collected on day 7 were assessed by flow cytometry, revealing comparable tumor cell siRNA uptake by si < (EG$_{18}$L)$_2$ and si < (EG$_{18}$L$_{diacid}$)$_2$ (Fig. 6g). These two conjugates also achieved similar levels of luciferase knockdown (Fig. 6h). In all, tumor delivery and activity of the diacid structure was similar to the parent si < (EG$_{18}$L)$_2$, despite the significantly increased albumin affinity of si < (EG$_{18}$L$_{diacid}$)$_2$. These data suggest that, for in vivo tumor gene silencing activity, the role of lipid hydrophobicity is of balanced importance with the value of increasing albumin affinity beyond the already high-affinity binding of si < (EG$_{18}$L)$_2$.

In a related set of studies, we also investigated the roles of the hydrophobic lipid beyond albumin binding by comparison of si < (EG$_{18}$L)$_2$ to siRNA duplexes directly conjugated to mouse serum albumin (MSA). The free thiols of MSA were modified with an azido-PEG$_3$-maleimide linker and then reacted with DBCO-modified siRNA duplex (Fig. 7a). EMSA revealed the expected mobility shift of resulting product relative to the DBCO-duplex precursor (Fig. 7b). Following i.v. delivery, nearly 80% of the siRNA delivered in the covalent MSA conjugate was found in the albumin-containing plasma fraction (Fig. 7c). Interestingly, the observed $t_{1/2}$ of the si-covalent-MSA was less than 10 min, far less than the >60 min seen for si < (EG$_{18}$L)$_2$ (Fig. 7d). Kidney accumulation of si-covalent-MSA was similar to what was seen for si < (EG$_{18}$L)$_2$ (Fig. 7e), suggesting that renal clearance is not the primary driver of diminished $t_{1/2}$ for si-covalent-MSA. It is possible that cell surface glycoproteins gp18 and gp30 bind to si-covalent-MSA, trafficking it for lysosomal degradation, as described in previous reports of modified albumin conjugates[61–63], rather than binding to gp60, which facilitates transcytosis. Alternatively, direct conjugation to siRNA may interfere with neonatal Fc receptor (FcRn) binding, the phenomenon which confers albumin with an exceptionally long half-life and counteracts cell internalization by recycling it to the extracellular space, facilitating ultimate return to the bloodstream.

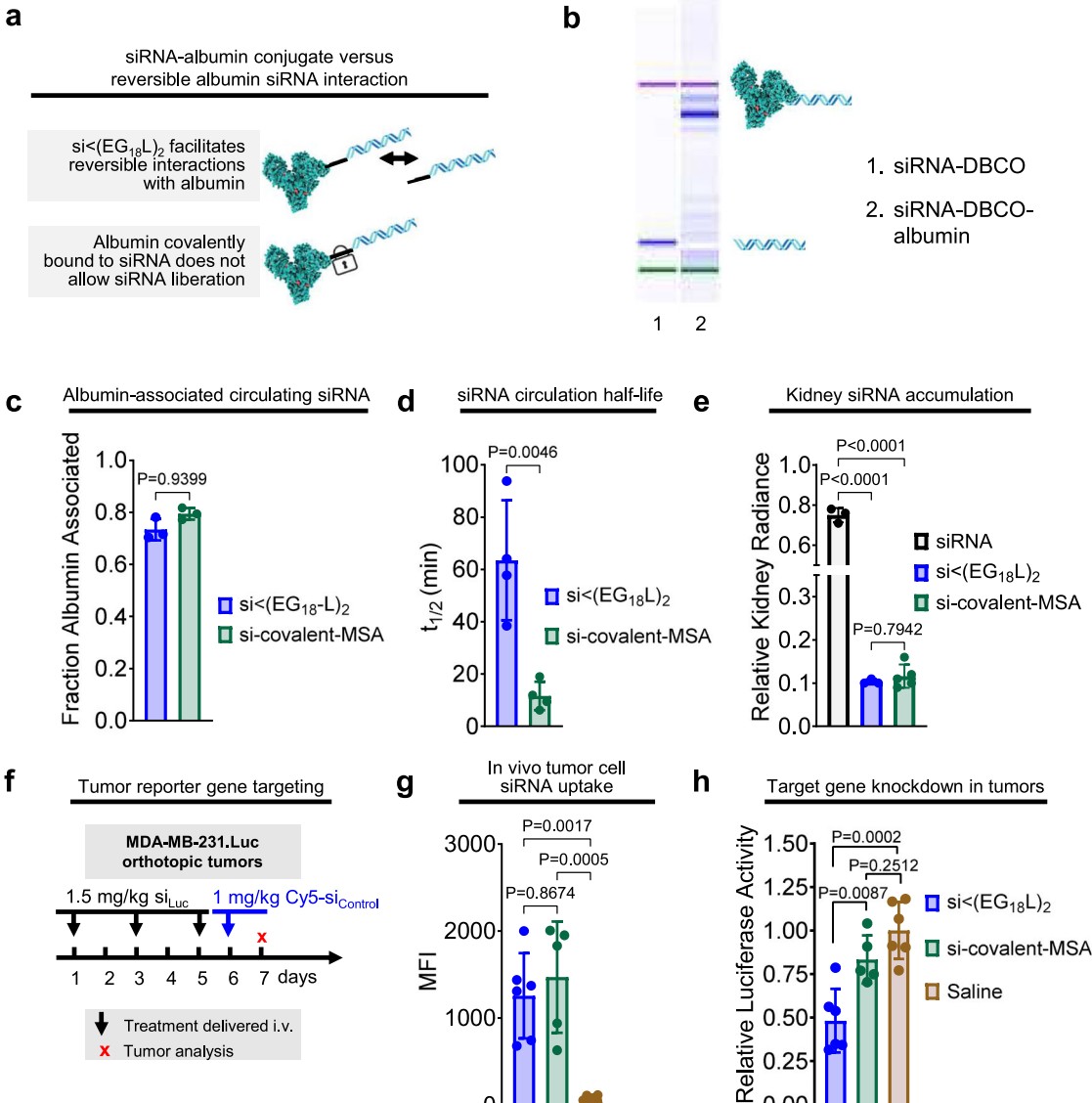

**Fig. 7 | Reversible siRNA interaction with albumin, enabled by siRNA-lipid conjugation, enhances siRNA circulation and in vivo bioactivity. a** Schematic model describing how siRNA conjugated to lipid enables reversible albumin interactions, as opposed to the irreversible nature of siRNA covalently linked with albumin. **b** Confirmation of conjugation of siRNA to albumin visualized using an Agilent 2100 Bioanalyzer. **c** Cy5-labeled siRNA-lipid conjugates or siRNA-albumin were delivered i.v. (1 mg/kg siRNA). Size exclusion chromatographic measurements of Cy5 fluorescence in albumin-containing plasma fraction ($n = 3$). **d** Intravascular Cy5 fluorescence measured through -1 h was used to calculate siRNA circulation half-life ($n = 4$). **e** Cy5 fluorescence in kidneys, lungs, liver, spleen, and heart was measured 1 h after treatment. Fluorescence values in kidneys relative to the total fluorescent values of all organs combined is shown ($n = 3–5$). **f** Schematic showing treatment schedule for i.v. delivery of siRNA-lipid conjugates to tumor-bearing mice. **g** Dissociated tumors collected on treatment day 7 were assessed for Cy5 fluorescence by flow cytometry or (**h**) lysed and assessed for luciferase activity ($n = 5–6$). For (**c–e, g, h**), each bar represents the average value (± S.D.), and each point represents values measured for each individual sample. Significance was assigned by one-way ANOVA with Tukey's multiple comparisons test.

Following delivery to tumor-bearing mice, IVIS and flow cytometry revealed comparable tumor delivery of the two compounds (Fig. 7f, g and Supplementary Fig. 14). However, the percentage reduction of tumor luciferase activity by $\mathrm{si_{Luc}}$-covalent-MSA was only approximately half of that created by $\mathrm{si_{Luc}} < (\mathrm{EG_{18}L})_2$ (Fig. 7h). These data further suggest that reversible albumin binding and presence of the lipid tails of $\mathrm{si} < (\mathrm{EG_{18}L})_2$ contributes to conjugate bioactivity.

The combined results from the diacid and direct albumin conjugate studies suggest that both reversibility of albumin binding and inclusion of a hydrophobic lipid to promote cell uptake are important features of the $\mathrm{si} < (\mathrm{EG_{18}L})_2$ design. Due to the different and somewhat unexpected circulation behavior of the direct conjugate, we sought to further probe this aspect of the design space by using $\mathrm{si} < (\mathrm{EG_{18}})_2$ lacking a lipid tail, and also using $\mathrm{si} < (\mathrm{EG_{18}L})_2$ harboring cleavability in the linker that would release the lipid tail; the linkers were designed to be either reducible through the presence of a disulfide [$\mathrm{si\text{-}(S\text{-}S)} < (\mathrm{EG_{18}L})_2$] or to be enzymatically degradable by inclusion of a nuclease-susceptible dithymidine [$\mathrm{si\text{-}(dT)_2} < (\mathrm{EG_{18}L})_2$] (Fig. 8a). When administered intravenously to tumor-bearing mice, omission of the lipid tail, as hypothesized, results in significantly higher kidney accumulation and lower tumor accumulation (Fig. 8b–d). While programmed release of the lipid tail resulted in comparable kidney clearance, diminished tumor gene silencing was observed in vivo across all degradable conjugates tested (Fig. 8e), further supporting the claim that presence of the intact lipid tail is critical for $\mathrm{si} < (\mathrm{EG_{18}L})_2$ activity.

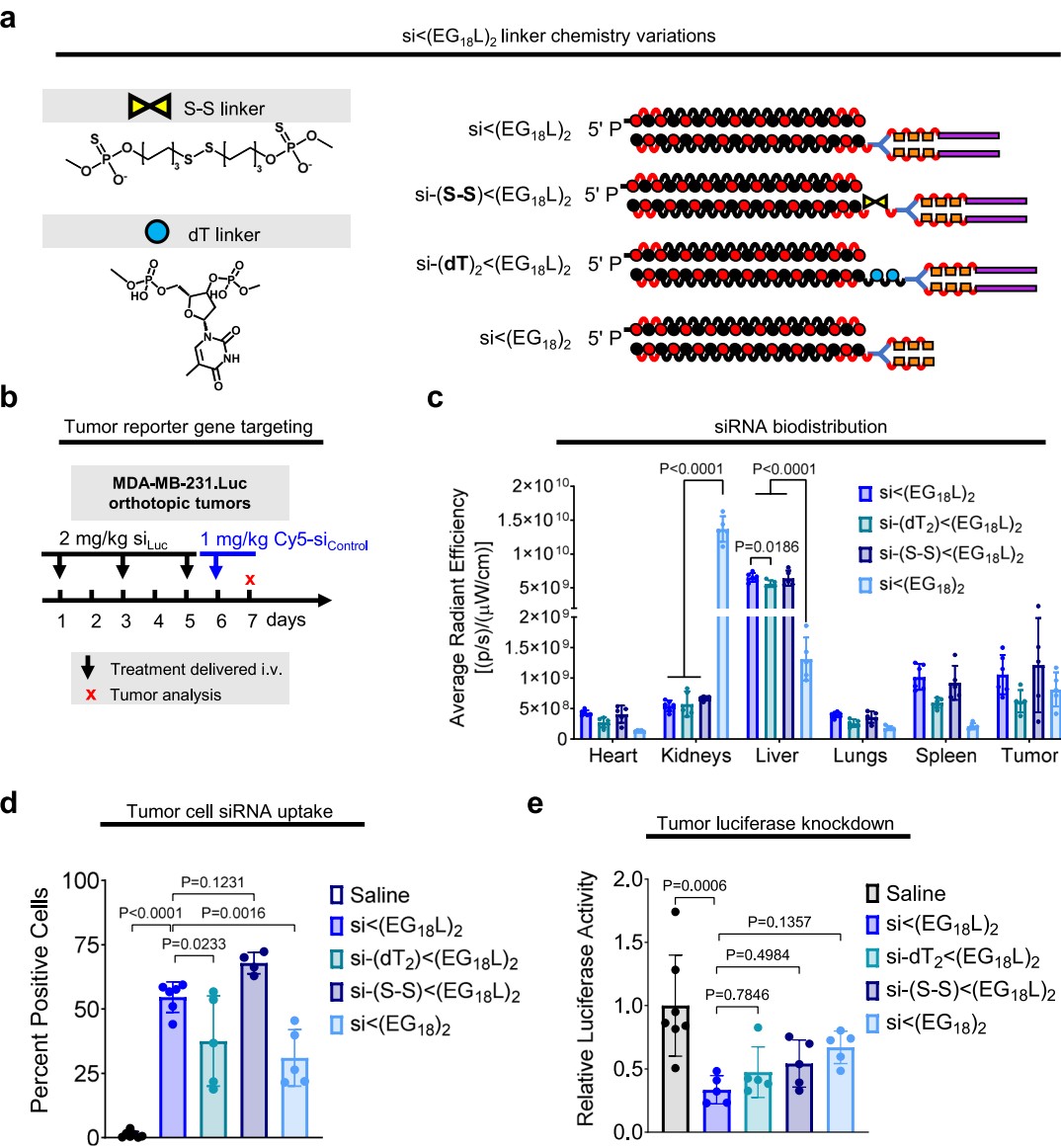

**Fig. 8 | Interrogation of siRNA conjugate lipid tail presence. a** Structures of the si < (EG$_{18}$L$_2$) variants to confirm the function of the lipids. **b** Treatment schematic showing i.v. dosing of conjugates to tumor-bearing mice. **c** Biodistribution of Cy5-labeled siRNA conjugates measured by epifluorescence 18 h after 1 mg/kg intravenous injection. Significance was assessed by two-way ANOVA with Tukey's multiple comparisons (n = 4–6). **d** Percent Cy5 positive tumor cells determined by flow cytometry. **e** Relative luciferase activity of tumor lysates after 3 × 2 mg/kg intravenous injections of siRNA conjugates targeting firefly luciferase over the course of one week. Significance assessed by one-way ANOVA with Tukey's multiple comparisons (n = 4–7). Significance compared to si<(EG$_{18}$L)$_2$ is displayed. Each bar represents the average value (± S.D.), and each point represents values measured for each independent sample.

## Mechanisms of si < (EG$_{18}$L)$_2$ cell internalization and trafficking

The combined si < (EG$_{18}$L$_{diacid}$)$_2$, albumin covalent conjugate, and degradable linker data sets led us to pose the hypothesis that the ideal siRNA structure will preferentially hitchhike on albumin in the bloodstream, but that this association should be reversible in order to allow the hydrophobic lipids to also participate in cell and endosomal membrane penetration. We anticipate that albumin-bound siRNAs can participate in cell internalization based on high tumor cell albumin uptake and catabolism[9] but that the efficacy of si < (EG$_{18}$L)$_2$ may be at least in part driven by its inherent hydrophobicity contributing to its ability to overcome cell-level barriers.

To elucidate cell internalization of conjugates, we first treated cells with either si < (EG$_{18}$L)$_2$ or si < (EG$_{18}$L$_{diacid}$)$_2$ in the absence of serum and in the presence of increasing excess of free albumin (Fig. 9a). This experiment revealed that the si < (EG$_{18}$L)$_2$ has inherently higher cell uptake than si < (EG$_{18}$L$_{diacid}$)$_2$ likely due to the charged and more hydrophilic diacids reducing cell membrane insertion. This study also revealed that increasing albumin concentrations correlated with reduced si < (EG$_{18}$L)$_2$ but not si < (EG$_{18}$L$_{diacid}$)$_2$ uptake. This result led us to postulate that albumin-bound siRNAs in general and si < (EG$_{18}$L$_{diacid}$)$_2$ (either bound or free form) enter the cell by a form of active endocytosis, likely pinocytosis, while si < (EG$_{18}$L)$_2$ has the inherent ability to interact or fuse with cell membranes through insertion of the hydrophobic stearate groups. To test this hypothesis, we treated cells with both conjugate structures either at 37 or 4 °C, with reduced temperature utilized to inhibit all endocytic processes. Strikingly, cell association of si < (EG$_{18}$L)$_2$ was not reduced by the 4°C condition, while it completely abrogated uptake of si < (EG$_{18}$L$_{diacid}$)$_2$ (Fig. 9b). These results indicate that si < (EG$_{18}$L)$_2$ has a unique combination of hydrophobicity and ideally tuned albumin affinity that contribute to its ability to be optimal for both albumin hitchhiking in circulation and overcoming cell-level barriers. To further gauge

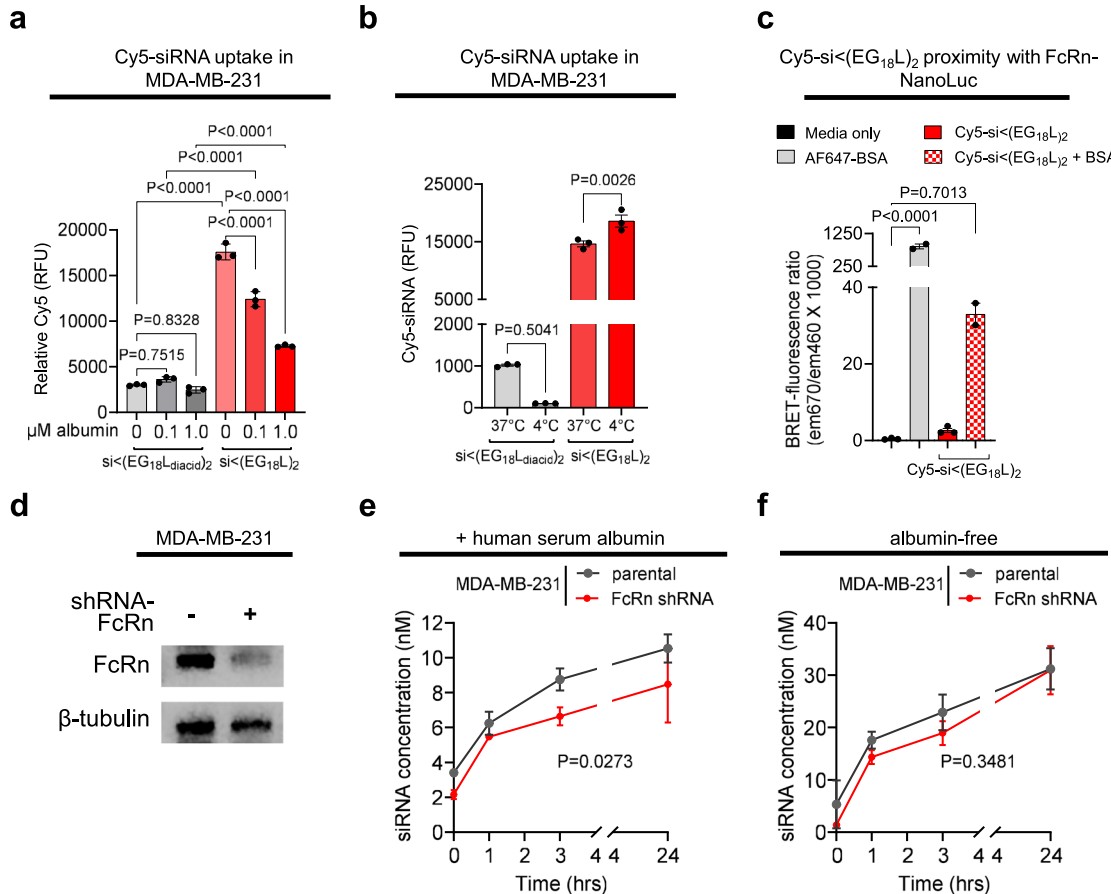

**Fig. 9 | si < (EG$_{18}$L)$_2$ uptake is driven by hydrophobicity and does not depend on active endocytic processes, while being partially susceptible to FcRn-mediated extracellular recycling when albumin bound. a** Cy5 fluorescence measured by flow cytometry of MDA-MB-231 cells incubated with 100 nM Cy5-labeled si < (EG$_{18}$L)$_2$ or si < (EG$_{18}$L$_{diacid}$)$_2$ in serum-free media, with or without albumin, for 4 h at 37 °C. **b** Cy5 fluorescence measured by flow cytometry of MDA-MB-231 cells incubated with 100 nM Cy5-labeled si < (EG$_{18}$L)$_2$ or si < (EG$_{18}$L$_{diacid}$)$_2$ in OptiMEM for 2 h at 37 °C vs. 4 °C. **c** HEK293 cells were transfected to express NanoLuc-tagged human FcRn and treated with media only, Alexa Fluor 647-BSA (positive control), or Cy5-labeled siRNA si < (EG$_{18}$L)$_2$ with or without BSA. BRET

signal (ratio of 670 nm vs. 460 nm signal) was measured. **d** Western blot of cell lysates collected from WT MDA-MB-231 or cells treated with shFcRn. Time course of siRNA < (EG$_{18}$L)$_2$ recycling in MDA-MB-231 parental vs. shFCGRT cells in the presence (**e**) or absence (**f**) of albumin, measured as siRNA concentration in media following 24 h incubation with 1 μM siRNA < (EG$_{18}$L)$_2$ with or without 1 μM unlabeled human serum albumin. Concentrations were adjusted for cell quantity using CellTiter Glo. For (**a**–**c**), significance was assigned by one way ANOVA with Sidak's multiple comparisons test (n = 2–3 experimental replicates). For (**d**, **e**) significance was assigned using an unpaired t-test on the AUC of the curves (n = 3 experimental replicates per timepoint).

whether si < (EG$_{18}$L)$_2$ was optimal with regard to divalent stearate presentation for overcoming cell and endosomal membrane barriers, we also tested the full, original library of divalent stearyl-modified conjugates and controls in the absence of albumin for their in vitro knockdown potency (Supplementary Fig. 15). This study revealed markedly similar trends in silencing potency as observed in vivo in Fig. 3c. These results suggest that si < (EG$_{18}$L)$_2$ is structurally optimized such that the divalent stearates can preferentially but reversibly bind to albumin while in the circulation, while also potentiating intracellular delivery after biodistribution to the target tissue.

We next examined intracellular trafficking of si < (EG$_{18}$L)$_2$ specifically assessing the role of the albumin-binding protein, neonatal Fc receptor (FcRn). The pH-dependent, high affinity interaction between albumin and FcRn within acidic endosomes promotes extracellular albumin recycling, prevents lysosomal albumin catabolism, and contributes to the long-half-life of albumin[64]. Overexpression of FcRn in human cancers has also been implicated in uptake and use of albumin to increase cellular growth[65] This may in turn facilitate uptake of albumin-associated payloads. The albumin-FcRn interaction is therefore potentially important in the context of intracellular-acting therapeutics that leverage albumin as a carrier, like si < (EG$_{18}$L)$_2$, in that it

may play a role in cellular uptake, cellular trafficking and recycling, and in vivo half-life.

We developed a cellular bioluminescence resonance energy transfer (BRET) assay[66] to measure albumin-mediated binding of si < (EG$_{18}$L)$_2$ to FcRn. We validated this cellular tool, wherein an FcRn-NanoLuciferase fusion protein (FcRn-NLuc) was expressed in HEK-293 cells, by treatment with albumin labeled with AF647, a fluorophore that can be excited by NLuc bioluminescence (Supplementary Fig. 16). This assay was then adapted to assess for binding between FcRn-NLuc and Cy5-labeled si < (EG$_{18}$L)$_2$, delivered in the presence or absence of albumin. These studies showed that BRET signal was produced by Cy5-si < (EG$_{18}$L)$_2$, but only in the presence of albumin, suggesting that albumin-bound si < (EG$_{18}$L)$_2$ interacts with FcRn within cells (Fig. 9c).

Knowing that Cy5-si < (EG$_{18}$L)$_2$ interacts with FcRn via albumin binding, we pursued complementary studies assessing FcRn-mediated extracellular recycling of internalized si < (EG$_{18}$L)$_2$. For these studies, we created an MDA-MB-231 cell line with stable FcRn knockdown (MDA231-shFcRn) (Fig. 9d and Supplementary Fig. 17). There was a modest but statistically significant decrease in albumin-dependent si < (EG$_{18}$L)$_2$ recycling in MDA231-shFcRn cells relative to parental MDA-MB-231 (Fig. 9e). However, si < (EG$_{18}$L)$_2$ recycling in the absence

of albumin was unaffected by FcRn knockdown (Fig. 9f). These data confirm that albumin-bound si < (EG$_{18}$L)$_2$ binds FcRn, which may impact its extracellular recycling. Although the serum half-life of mouse serum albumin is ~35 h[67], the absolute half-life of our optimized conjugate is fewer than 2 h. These combined data imply that albumin-mediated interaction of si < (EG$_{18}$L)$_2$ with FcRn has only a modest impact on cellular recycling and, consequently, intracellular delivery or plasma pharmacokinetics.

Finally, we sought to further examine the intracellular trafficking of si < (EG$_{18}$L)$_2$ with regard to endosomal escape. Using a Gal8 reporter assay in MDA-MB-231 cells to visualize endosomal disruption[68], we noted that, in contrast to cationic transfection agents, si < (EG$_{18}$L)$_2$ conjugates did not produce Gal8 foci indicative of endosomal disruption (Supplementary Fig. 18). Nevertheless, after 24 h of incubation with si < (EG$_{18}$L)$_2$ (Supplementary Fig. 18), we observed cytoplasmic distribution of the conjugate, consistent with our data that confirm conjugate silencing function. The lipid conjugates may facilitate fusion and gradual penetration of endosomal membranes without causing membrane disruption. The leak of bioactive conjugates out of endosomes in the absence of their disruption is also consistent with other observations that the endolysosomal system can serve as an intracellular depot for siRNAs that are chemically stabilized to remain intact within these degradative vesicles[69].

## Albumin-binding siRNA conjugate shows greater efficacy for TNBC therapy than a small molecule inhibitor of MCL-1

To assess the utility of si < (EG$_{18}$L)$_2$ for gene targeting in TNBC, HCC70 human TNBC cells were treated with si < (EG$_{18}$L)$_2$ structures based on siRNA sequences directed against *MCL1* (si$_{MCL1}$-L$_2$). This system achieved carrier-free, dose-dependent *MCL1* knockdown (Fig. 10a) and induced caspase 3/7 activity (Fig. 10b), a hallmark of the intrinsic apoptosis pathway and consistent with the phenotypic consequences of MCL-1 loss of function. Similar results were found in HCC1187 and MDA-MB-231 cells (Supplementary Fig. 19). Orthotopic HCC70 tumors grown in female athymic mice were treated with si$_{MCL1}$-L$_2$ (10 mg/kg), resulting in potent MCL-1 downregulation at treatment day 8, as measured by IHC (Fig. 10c), correlating with induction of caspase-3 cleavage (Fig. 10d). In parallel, mice bearing HCC70 tumors were treated with 12.5 mg/kg of the MCL-1 small molecule inhibitor MIK665 developed by Servier and Novartis; this matches the maximum tolerated dose of this compound[70] and is more than 18 times the molar quantity of si$_{MCL1}$-L$_2$ delivered. However, MIK665 induced caspase 3 cleavage to a lesser extent and reduced cell proliferation to a lower degree than what was seen for si$_{MCL1}$-L$_2$. (Fig. 10d). Similar results were seen in HCC1187 and MDA-MB-231 tumors at treatment day 8 (Supplementary Fig. 20).

In longer term efficacy studies, weekly si$_{MCL1}$-L$_2$ treatment through day 28 inhibited growth in 8/8 HCC70 tumors, 4/6 HCC1187 tumors, and 4/10 MDA-MB-231 as compared to si$_{Control}$-L$_2$, leading to overall greater survival (Fig. 10e, f). Interestingly, treatment with MIK665 did not alter tumor growth in any MDA-MB-231 tumors, and just 1/7 HCC70 tumors had diminished growth with the small molecule treatment (Supplementary Fig. 21). These data reveal a potential therapeutic advantage of *MCL1* knockdown over small molecule MCL-1 inhibition in this setting. Though the reasons underlying these results are uncertain, it is often observed that tumors respond to small molecule MCL-1 inhibition by rapidly upregulating MCL-1 protein[71–75] translation, potentially increasing resistance to MCL-1 inhibition[76]. Importantly, this scenario would be precluded by targeted knockdown of *MCL1* transcripts. Further, MCL-1 may have functions unrelated to its BH3-binding domain, which might be carried out unabated by BH3 mimetics like MIK665 but lost upon *MCL1* knockdown[77–79]. Importantly, no significant toxicities were observed in mice treated with si$_{MCL1}$-L$_2$, even at the highest dose tested (20 mg/kg), as measured by blood chemistry, complete blood count (CBC), and B cells isolated from the bone marrow and spleen (Supplementary Figs. 8 and 9). In contrast, toxicities often observed in response to MCL-1 small molecule inhibitors include B-cell and erythrocyte depletion, hemolysis, and weight loss[70]. Toxicity readouts were not measured here for MIK665 treated mice, as this small molecule has much higher affinity for human over mouse MCL1 protein; therefore, mouse studies are not representative of the toxicity profile[70].

This work shows the optimization of carrier-free, albumin-mediated tumor delivery and bioactivity of siRNA in vivo, through systematic variation of lipid-siRNA conjugate valency, linker length, phosphorothioate bonds, lipid chemistry, linker degradability, and linker branching architecture. Modest dosing with this construct achieved nearly 80% carrier-free knockdown in an orthotopic tumor model of triple negative breast cancer, a significant advancement for carrier-free, extra-hepatic RNAi delivery.

In assessing the impact of a hydrophilic linker on albumin binding, we identified an optimal length [si < (EG$_{18}$L)$_2$] and branch point placement of the divalent lipid. Notably, designs with the hydrophilic linker before the branching point of the divalent structure, when matched for both overall hydrophobicity or length between the siRNA and lipids, showed inferior albumin association in plasma in vivo, which may be attributable to greater lipoprotein association and propensity for self-assembly (lower CMC). Our studies also showed that phosphorothioate, rather than phosphodiester, linkages are beneficial within the linker structure and at the sense strand terminus where the albumin-binding moiety is located, likely contributing to plasma albumin association.

Notably, a combination of selective, but reversible albumin binding in the circulation, combined with an optimal level of hydrophobicity to promote cell uptake whenever the conjugate dissociates from albumin after reaching the tumor site, is critical to the overall performance of si < (EG$_{18}$L)$_2$. To further support this point, it was shown that direct albumin conjugation, diacid conjugates with higher albumin binding affinity and lower cell membrane association, and conjugates with a degradable linker that can release the C$_{18}$ moieties all have reduced tumor gene silencing function. Relatedly, it was discerned that FcRn binding of si < (EG$_{18}$L)$_2$ via its association with albumin has a significant but relatively small magnitude impact on its intracellular recycling, further suggesting that the lipids and not association with albumin dictate trafficking at the cellular level once the conjugate reaches the target tissue site.

Finally, si < (EG$_{18}$L)$_2$ was shown to provide significant knockdown and tumor growth inhibition in vivo through silencing of oncogenic *MCL1* in three models of TNBC; the siRNA was also found to outperform a clinically tested MCL-1 small molecule inhibitor given at its MTD on the same treatment schedule. Of critical importance, little *MCL1* knockdown was observed in livers of treated mice, and no toxicities were observed with control or *MCL1*-directed sequences.

Overall, this work has important implications for the delivery of siRNAs to extrahepatic targets, a goal that has remained clinically elusive. Albumin is known to accumulate at sites of inflammation and vascular leakiness, which are associated with a wide array of diseases. Therefore, insights gleaned from si < (EG$_{18}$L)$_2$ development for in situ albumin binding, carrier-free delivery will support the discovery of extrahepatic, carrier-free siRNA therapeutics in cancer treatment and beyond.

## Methods

All animal experiments were IACUC approved and performed according to AAALAC guidelines and NIH best practices in AAALAC-accredited facilities at Vanderbilt University.

### Reagents

2'-O-Me and 2'-F phosphoramidites, universal synthesis columns (MM1-2500-1), and all ancillary RNA synthesis reagents were purchased

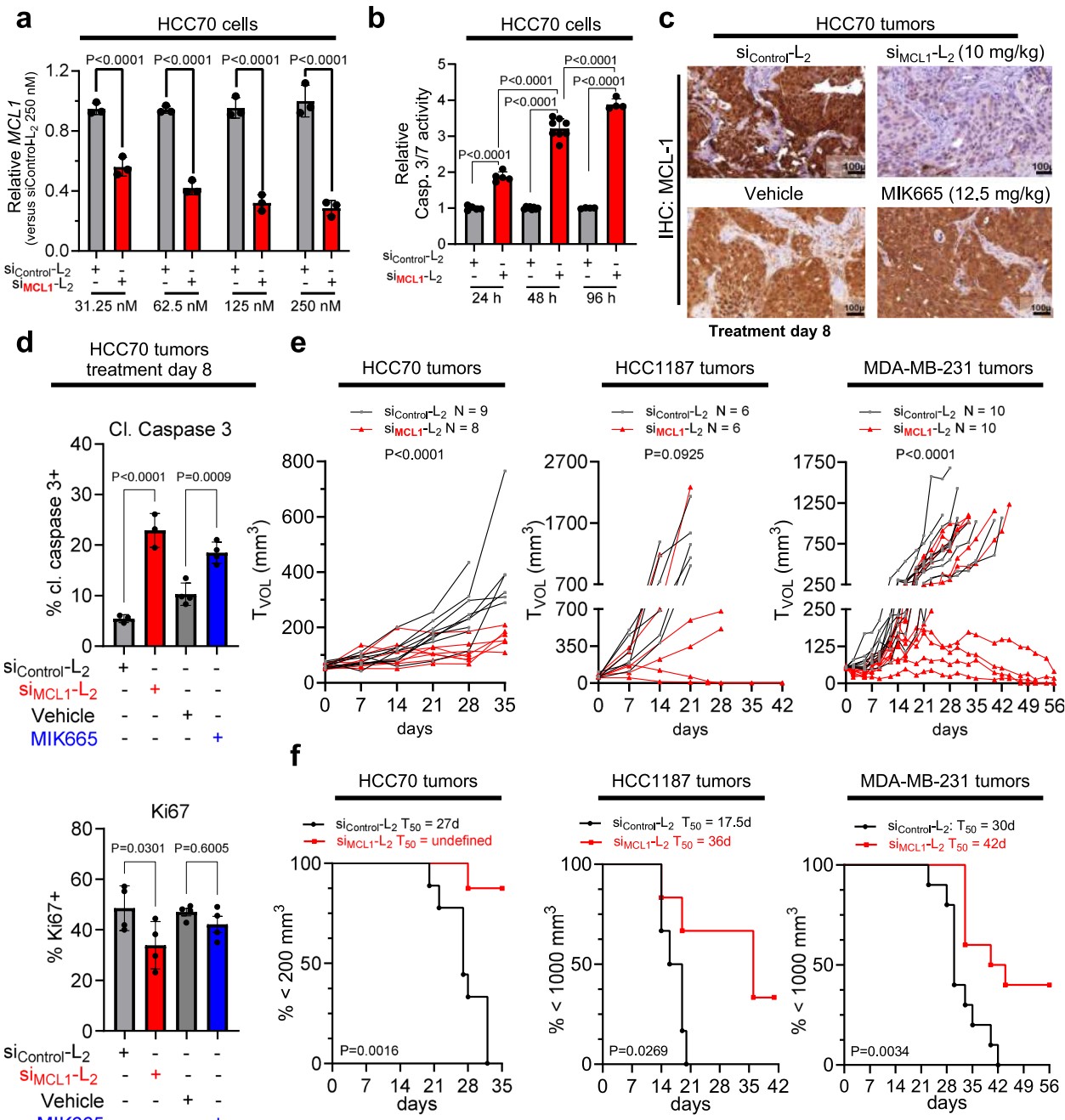

**Fig. 10 | Carrier-free delivery of MCL1-targeting siRNA diminishes MCL-1 expression, tumor cell survival, and tumor growth in TNBC. a** HCC70 cells were treated in serum-free OptiMEM with siRNA-lipid conjugates and assessed 48 h post-treatment for expression of *MCL1* transcript levels relative to the housekeeping gene *PPIB*. Data are shown relative to *MCL1* levels seen in $si_{Control}$-$L_2$ (250 nM)-treated cells ($n = 3$). **b** HCC70 cells treated with siRNA-lipid conjugates for 24–96 h at 100 nM were assessed using Caspase 3/7-Glo ($n = 4$–8). **c, d** Mice bearing orthotopic HCC70 tumors were treated on day 0 and day 7 with $si_{Control}$-$L_2$ or $si_{MCL1}$-$L_2$ (10 mg/kg, delivered i.v.). Tumors were collected on day 8. $n = 3$–4. **c** Representative images of HCC70 tumors assessed by IHC for MCL-1. **d** Quantification of the percentage of HCC70 tumor cells staining positive for Ki67 and cleaved (cl.) caspase 3 ($n = 3$–4). **e, f** Mice harboring HCC70 ($n = 8$–9), HCC1187 ($n = 6$), and MDA-MB-231 ($n = 8$–10) orthotopic tumors (50–100 mm$^3$) received once weekly i.v. treatment

with $si_{Control}$-$L_2$, $si_{MCL1}$-$L_2$, vehicle (used for delivery of MIK665), or MIK665 through day 28 (HCC70 and HCC1187) and through day 49 (MDA-MB-231). Tumors were collected on day 8 (HCC70), day 35 (HCC70), day 42 (HCC1187), and day 56 (MDA-MB-231) or earlier if tumors ulcerated or exceeded humane size limitations. **e** Tumor volumes were measured throughout treatment. Statistical values were calculated based on area under the curve. **f** Kaplan–Meier analysis of tumor-bearing mice, defining survival as tumor volume under 1000 mm$^3$ (HCC1187 and MDA-MB-231) or 200 mm$^3$ (HCC70, which exhibited slow tumor growth rate). Average time to exceed the defined tumor volume ($T_{50}$) for each group is shown above each panel. *p* values are calculated using the log-rank (Mantel–Cox) test. For (**a, b**), significance was assigned by two-way ANOVA with Sidak's multiple comparisons test. For (**d**), significance was assigned using one-way ANOVA with Tukey's multiple comparisons test. For (**e**), significance was determined using an unpaired *t*-test.

from Bioautomation (Irving, TX). Symmetrical branching 2′-cyanoethyl, diisopropyl (CED) phosphoramidite (CLP-5215) was obtained from ChemGenes (Wilmington, MA). Cyanine 5 (Cy5) phosphoramidite (10-5915), stearyl phosphoramidite (10-1979), biotin

triethyleneglycolyl (TEG) phosphoramidite (10-1955), hexaethyleneglycol phosphoramidite (10-1918), TEG cholesterol phosphoramidite (10-1976), 5′-Amino-Modifier 5 (10-1905), and desalting columns (60-5010) were all purchased from Glen Research (Sterling,

VA). PE (phycoerythrin) anti-mouse CD19 antibody (Catalog No. 115508) and APC (allophycocyanin) anti-mouse/human CD45R/B220 antibody (Catalog No. 103212) were acquired from BioLegend (San Diego, CA). MIK665 was from Selleck Chem (Houston, TX). All other reagents were purchased from Sigma-Aldrich unless otherwise specified.

## Conjugate synthesis, purification, and validation

Oligonucleotides were synthesized using 2′-F and 2′-O-Me phosphoramidites with standard protecting groups on a MerMade 12 Oligonucleotide Synthesizer from Bioautomation (Irving, TX). siRNA sequences can be found in Supplementary Table 1. Amidites were dissolved at 0.1 M in anhydrous acetonitrile, and 2′OMe U-CE phosphoramidite used 20% anhydrous dimethylformamide as a cosolvent. Stearyl phosphoramidite was dissolved in 3:1 (v:v) dichloromethane:acetonitrile. Coupling was performed under standard conditions, and strands were grown on controlled pore glass with a universal terminus (1 μmol scale, 1000 Å pore).

Strands were cleaved and deprotected using 1:1 methylamine:40% ammonium hydroxide (25 °C, 2 h). Lipophilic RNAs were purified by reversed-phase high performance liquid chromatography (HPLC) using a Clarity Oligo-RP column from Phenomenex (Torrance, CA) under a linear gradient [85% mobile phase A (50 mM triethylammonium acetate in water) to 100% mobile phase B (methanol) or 95% mobile phase A to 100% mobile phase B (acetonitrile)]. Oligonucleotide fractions were dried (Savant SpeedVac SPD 120 Vacuum Concentrator, Thermo Fisher), resuspended in nuclease free water, sterile filtered, and lyophilized.

Conjugate molecular weight and purity (Supplementary Fig. 1) was confirmed using liquid chromatography-mass spectrometry (LC-MS, Thermo Fisher LTQ Orbitrap XL Linear Ion Trap Mass Spectrometer). Chromatography was performed using a Waters XBridge Oligonucleotide BEH C18 Column under a linear gradient [85% phase A (16.3 mM triethylamine–400 mM hexafluoroisopropanol) to 100% phase B (methanol)] at 45 °C. Control conjugate, si-PEG$_{45}$ < L$_2$, molecular weight was validated using MALDI-TOF MS as previously reported[40] using 50 mg/ml 3-hydroxypicolinic acid in 1:1 (v:v) water:acetonitrile with 5 mg/ml ammonium citrate as a matrix. Synthesis of amine-reactive lipids and subsequent modification of oligonucleotides to make unsaturated and diacid lipid conjugates were adapted from methods reported by Prakash et al.[36] and described in detail in the Supplementary Methods. Purified oligonucleotide was resuspended in 0.9% sterile NaCl and annealed to its complementary strand by heating to 95 °C and cooling at a rate of 1.66 °C/min until 25 °C. Cyanine-5 fluorophore was incorporated on the opposite end of the duplex (5′ antisense) as the lipophilic moiety (5′ sense) in order to avoid contributing to lipid-driven micellization. Duplexes covalently conjugated to albumin were synthesized in a two-step, one-pot reaction described in detail in the Supplementary Methods. Fluorophore-labeled duplex was used to cross-validate A260 concentration measurements using fluorescence.

## Cell culture

Cells were cultured in Dulbecco's modified eagle's medium (DMEM, Gibco), containing 4.5 g/l glucose, 10% fetal bovine serum [FBS (Gibco)], and 50 μg/ml gentamicin. *Mycoplasma* contamination testing was done by MycoAlert Mycoplasma Detection Kit from Lonza (Lexington, MA). MDA-MB-231 (CRM-HTB-26), HCC1187 (CRL-2322), and HCC70 (CRL-2315) were purchased from ATCC (Manassas, VA). MDA-MB-231 cells stably expressing a bicistronic green fluorescent protein (GFP) and firefly luciferase (Luc) expression vector (MDA-MB-231.Luc) were seeded at 4000/well in 96-well plates. HCC70 and HCC1187 cells were seeded at 10,000/well in 96-well plates. After 24 h, cells were treated with siRNA (25 nM) using Lipofectamine 2000 (Thermo Fisher) in OptiMEM according to the manufacturer's protocol or treated

carrier-free with si < (EG$_{18}$L)$_2$ (31.25 nM–250 nM dose range), replacing with complete media at 24 h post-transfection. Luciferase activity was measured at 48 h post-transfection in cells treated for 5 min with 150 μg/ml D-Luciferin potassium salt (Thermo Fisher) using an IVIS Lumina III imaging system from Caliper Life Sciences (Waltham, MA). Cells were then assessed for caspase activity using Caspase 3/7-Glo at 24 h intervals after adding siRNA conjugate. Caspase-3/7 activity was measured 96 h after treatment using Caspase 3/7-Glo from Promega (Madison, WI) according to the manufacturer's directions. Total RNA was harvested at 96 h after treatment using an RNeasy kit from Qiagen (Hilden, Germany). *MCL1* and *PPIB* (housekeeping) mRNA expression was measured in 0.5 μg purified RNA using Quantigene assay (Thermo Fisher).

## In vitro uptake

For assessment of albumin dependence of uptake, MDA-MB-231 cells were plated at 10,000/well in a 96-well plate 24 h prior to treatment with 100 nM of Cy5-labeled si < (EG$_{18}$L)$_2$ or si < (EG$_{18}$L$_{diacid}$)$_2$, in Opti-MEM alone (Gibco) or premixed with human serum albumin (Sigma) at 100 nM or 1 μM final concentration in Opti-MEM. The treatment duration was 4 h at 37 °C. Cells were washed with PBS, trypsinized, and suspended in PBS + 2% FBS. Fluorescent uptake was assessed using a Guava easyCyte HT flow cytometer (Luminex) with data analyzed in FlowJo (BD Biosciences). For assessment of temperature dependence of uptake, MDA-MB-231 cells were plated as above and treated with 100 nM of Cy5-labeled si < (EG$_{18}$L)$_2$ or si < (EG$_{18}$L$_{diacid}$)$_2$ in Opti-MEM (Gibco) for 2 h at 37 °C vs. 4 °C (*n* = 3 each). Cell processing and flow cytometry analysis were performed as above.

## Serum stability

siRNA (0.1 nmol) in 60% FBS in PBS was incubated at 37° for 0–48 h, then resolved on a 2% agarose gel in 1X TAE Buffer. Gels were stained with GelRed Nucleic Acid Stain from Biotium (Fremont, CA) according to the manufacturer's protocol and imaged with UV transillumination.

## Immunofluorescence

Samples were snap frozen in OCT embedding medium. Cryosections (6 μm) at multiple tissue depths were fixed for 10 min in 4% paraformaldehyde (PFA) and then stained with rabbit anti-firefly luciferase antibody [anti-Fluc; 1:500; ab185924, Abcam (Waltham, MA)] and goat anti-rabbit Alexa Fluor® 488 [1:500; ab150077, Abcam (Waltham, MA)]. Slides were then counterstained with DAPI and imaged on a Nikon Eclipse Ti inverted confocal microscope. Imaging settings were kept constant across different treatment groups.

## Biolayer interferometry

Binding kinetics were measured by biolayer interferometry (BLI) at 30 °C, 1000 rpm using an Octet RED 96 from ForteBio (Dallas, TX). Duplexes were synthesized with antisense strand 5′ terminal TEG-Biotin, diluted to 500 nM in Dulbecco's PBS containing Ca$^{2+}$ and Mg$^{2+}$ (DPBS$^{+/+}$), and loaded for 600 s on a Streptavidin Dip and Read Biosensor from ForteBio (Dallas, TX). Biosensor association to human and mouse albumin in DPBS$^{+/+}$ was measured for 300 s, followed by measurements of dissociation for 300 s. The binding values were measured using Octet Data Analysis HT Software. Interstep correction was performed by aligning to the dissociation step, and noise filtering was performed. Global analysis was performed to derive constants simultaneously from all tested analyte concentrations.

## Critical micelle concentration (CMC)

A duplex serial dilution was prepared in a 96-well plate from 20 μM to 10 nM in 50 μl of Ca$^{2+}$/Mg$^{2+}$ free DPBS with Nile Red (0.25 μg), and the plate was agitated at 37 °C in the dark for 2 h. Fluorescence was measured on a plate fluorimeter from Tecan (Mannedorf, Switzerland) at excitation 535 ± 10 nm and emission 612 ± 10 nm. The CMC was

defined, as previously described[80], as the intersection point on the plot of the two linear regions of the Nile Red fluorescence versus duplex concentration.

## Electrophoretic mobility shift assay

siRNA conjugates (0.1 nmol) were incubated with 5X molar excess of human or mouse albumin 30 min at 37 °C. Complexes were resolved on 4%–20% Mini-Protean TGX polyacrylamide gels from Bio-Rad Laboratories (Hercules, CA). Nucleic acid was visualized with GelRed Nucleic Acid Stain from Biotium (Fremont California) under ultraviolet imaging and protein visualized by Coomassie Blue stain under visible light imaging.

Conjugation efficacy of DBCO-modified siRNA duplex with azide-modified albumin was visualized using the Agilent Protein 230 Assay on the Agilent 2100 Bioanalyzer according to the manufacturer's instructions.

## Circulation half-life and biodistribution

Intravital fluorescence microscopy was performed using previously reported methods[81] on a Nikon Czsi+ system. Briefly, isoflurane-anesthetized, 6–8-week-old male CD-1 mice from Charles River (Wilmington, MA) were immobilized on a heated confocal microscope stage. Mouse ears were depilated and immobilized under a glass coverslip with microscope immersion fluid. Using light microscopy to visualize ear vessels, images were focused to the plane of greatest vessel width, where flowing red blood cells were visible. Confocal laser microscopy was used to acquire one image per second, at which point Cy5-labeled siRNA (1 mg/kg) was delivered via tail vein. Fluorescent intensity within a circular region of interest (ROI), drawn in the focused vein, was used to measure fluorescence decay. Values were normalized to maximum initial fluorescence and fit to a one-compartment model in PK Solver. For assessment of circulation half-life at time points >30 min, blood was collected at various time points from 5 min to 24 h in EDTA-coated tubes, diluted in sterile saline, and Cy5 fluorescence measured in 96-well plates by fluorimetry (Tecan). Cy5 fluorescence was quantified in whole organs (heart, lung, liver, kidney, and spleen) using an IVIS Lumina Imaging system from Caliper Life Sciences (Waltham, MA) at excitation and emission wavelengths of 620 and 670 nm, respectively, using Living Image software version 4.4.

## Size exclusion chromatography (SEC)

Murine plasma was collected 45 min after a 1 mg/kg intravenous injection of Cy5-labeled siRNA conjugate into 6–8 week old, male CD-1 mice from Charles River (Wilmington, MA). Plasma was filtered (0.22 μm) and then injected into an AKTA Pure Chromatography System from Cytiva (Marlborough, MA) with three inline Superdex 200 Increase columns (10/300 GL). Fractionation was done at 0.3 ml/min using Tris running buffer (10 mM Tris-HCl, 0.15 M NaCl, 0.2% NaN₃) into 1.5 ml fractions with a F9-C 96-well plate fraction collector (Cytiva). Cy5 fluorescence was measured in fractions (100 μl) in black, clear-bottom, 96-well plates from Greiner-Bio-One (Kremsmunster, Austria, REF 675096) on a SynergyMx from Biotek (Winooski, VT) at a gain of 120, excitation 642/9.0, emission 675/9.0. Fraction albumin-bound conjugate was determined by taking the sum of fluorescence intensity for fractions associated with albumin elution divided by the sum of fluorescence intensity for all fractions collected. Albumin-associated fractions were determined by running known protein standards through the SEC system and examining A280 of eluate from each of the fractions.

## Orthotopic mammary tumor studies

MDA-MB-231, MDA-MB-231.Luc, HCC70, or HCC1187 cells (1 × 10⁶) in 100 μl of 50% (v/v) Matrigel were injected into the inguinal mammary fat pads of 4–6-week-old female athymic (*nu/nu*) mice from Envigo (Indianapolis, IN). Mice were randomized into treatment groups when tumor volume reached 50 mm³ as measured by calipers using the formula $T_{vol} = (length × width^2) / 2$. Mice were treated by intravenous (i.v.) delivery at the indicated doses, durations, and frequencies. Conjugates were delivered in 0.9% saline. MIK665 treatments were formulated in 2% vitamin E/d-α-tocopheryl polyethylene glycol 1000 succinate (Sigma-Aldrich) in NaCl 0.9% (wt/vol) and delivered i.v. via tail vein at 12.5 mg/kg using the indicated schedule. Mice were humanely euthanized at treatment day 4, 8, 28, or 35 as indicated, or when tumors exceeded 1000 mm³. Tissues and tumors were collected at necropsy, and the samples were frozen for gene expression analysis or processed for other readouts as described below.

## Flow cytometry

For tumor cell uptake of Cy5-labeled siRNA-lipid conjugates, tumors were minced in HBSS (with Ca²⁺ and Mg²⁺), dissociated at 37 °C for 1 h in 0.5 mg/ml collagenase (Millipore Sigma)/0.19 mg/ml DNase (Bio-Rad) in DMEM, and filtered through a 70 μm cell strainer. Erythrocytes were removed using ACK lysis buffer (Thermo Fisher) for 2 min. Dissociated tumor cells in PBS⁻/⁻ were assessed by flow cytometry on a Guava easyCyte from Cytek (Fremont, CA) using FlowJo software. Cell populations were isolated using forward and side scatter. Gating was done to select for GFP+ tumor cells, and Cy5 fluorescence intensity was measured in this population.

For B cell analyses, spleens were minced in PBS⁻/⁻ and 1 mM EDTA, passed through a 70 μM cell strainer, and washed. Excised femorae and tibiae were flushed with 10 ml of 1% fetal calf serum in EDTA (5 mM) + PBS⁻/⁻ using a 25G needle. The exudate was filtered (70 μm) and washed. Erythrocytes were removed from splenocytes and bone marrow cells (BMCs) using ACK lysis buffer for 3 min. In total, 1 × 10⁶ cells/ml were blocked with mouse Fc block and then stained with the fluorophore-conjugated antibodies CD19-PE (1:200) and B200-APC (1:400). Samples were run on a BD FACS Diva. Fluorescence minus one and compensation controls were used to determine flow cytometer settings and gating schemes. B cells were defined as high intensity staining of CD19 PE and B200 APC.

## Gene expression analysis in tissues

For analysis of luciferase reporter silencing, tumor fragments (200–300 mg) were lysed for 1 h on ice with agitation in 1X Reporter Cell Lysis Buffer from Promega (Madison, WI) and then centrifuged at 14,000 × g for 15 min at 4 °C. Protein concentration was then quantitated using a Pierce BCA Assay (Thermo Fisher). Lysates (20 mg per well) were assessed in 96-well plates using 90 μl reconstituted Luciferase Assay Substrate from Promega (Madison, WI) according to the manufacturer's directions. Luminescence was measured using IVIS grid quantitation.

*MCL1* mRNA was measured in tumor and liver tissue using QuantiGene SinglePlex assay (Thermo Fisher). Tissues were harvested, stored, and homogenized in RNAlater (Thermo Fisher) at 4 °C. Samples were washed twice with water and then digested for 6 h at 55 °C in Quantigene Diluted Lysis Mixture (DLM) supplemented with proteinase K (0.25 mg/ml). Tissue lysates were diluted 1:2 for Quantigene assessment with manufacturer-designed probe sets directed against human (for tumor analysis) and mouse (for liver analysis) *MCL1* and *PPIB*. Luminescent signals generated from each specific probe set were measured and quantified on a plate luminometer from Tecan (Mannedorf, Switzerland). Each sample was assessed in 5 technical replicates. Values shown are the average *MCL1* (corrected for the loading control, *PPIB*), relative to the average *MCL1* value in tumors from saline-treated mice.

## Blood chemistry and complete blood count

Whole blood was collected in EDTA-coated tubes or spun down at 2000 × g for 15 min at 4 °C to isolate plasma. Samples were then

submitted to the Vanderbilt Translational Pathology Shared Resource for complete blood count and chemistry analyses.

## Generation of shFcRn MDA-MB-231 cell lines

FcRn shRNA lentiviral particles (sc-45632-SH) were purchased from Santa Cruz Biotechnology (Dallas, TX) and used to transduce MDA-MB-231 cells according to manufacturer instructions. Cells were selected for 2 weeks using 5 μg/ml puromycin and confirmed by Western blot for target protein expression.

## Label-free measurement of siRNA cellular recycling with and without FcRn silencing

Recycling of siRNAs out of the cell following internalization was measured using a label-free HPLC assay based on fluorescently labeled peptide nucleic acid (PNA) probes. This protocol was derived from an established method for siRNA in vivo pharmacokinetics measurement[82]. A 12-well plate was coated with 50 μg/ml rat tail collagen I (Gibco) in 20 mM acetic acid in $H_2O$ for 1 h at 37 °C and then washed with PBS. Wild-type MDA-MB-231 cells and shFcRn MDA-MB-231 cells with silenced FcRn were plated at 50,000 cells per well and incubated with high-glucose DMEM (Gibco) + 10% FBS (Corning) + pen/strep for 24 h. For albumin-free conditions ($n = 3$ for each cell line), cell media was exchanged for Opti-MEM (Gibco) for 1 h, then replaced with 1 μM $si_{Luc} < (EG_{18}L)_2$ in Opti-MEM for 24 h, followed by a PBS (Gibco) wash and replacement with fresh, serum-free Opti-MEM. For albumin-containing conditions ($n = 3$ for each cell line), cell media was replaced with 1 μM human serum albumin (Millipore Sigma) plus 1 μM $si_{Luc} < (EG_{18}L)_2$ in Opti-MEM for 24 h. This treatment was followed by a PBS wash and finally replacement with DMEM + 10% FBS + pen/strep. In all cases, 100 μl of media were withdrawn from each well immediately after media replacement ($t = 0$ h) and at 1 h, 3 h, and 24 h. Media was replenished to maintain consistent volume, with corresponding correction for dilutional effects as appropriate. Following sample collection, CellTiter Glo (Promega) was performed according to manufacturer protocol and quantified by luminometer (Tecan Infinite M1000). Media samples were immediately centrifuged at $1000 \times g$ for 5 min to remove any suspended cells or debris, and 100 μl of media supernatant was retained and stored at −20 °C. Standards were prepared from a serial dilution (156 fmol to 10 pmol) of $si_{Luc} < (EG_{18}L)_2$ in 100 μl Opti-MEM. To the thawed supernatant and standards, 100 μl of QuantiGene homogenization buffer (Invitrogen) plus 0.5 mg/ml Proteinase K (Invitrogen) was added, followed by 20 μl of 3 M KCl to precipitate SDS. Samples were centrifuged at $4000 \times g$ for 15 min, and supernatant was transferred to a fresh tube and centrifuged at $4000 \times g$ for 5 min. In total, 150 μl of this second supernatant was transferred to a fresh tube. To the samples, 100 μl of hybridization buffer (50 mM Tris in 9:1 water:acetonitrile, pH 8.8) and 2 μl of complementary PNA probe (PNA Bio) were added. Samples were vortexed and pulsed in a centrifuge, incubated at 90 °C for 15 min, and finally annealed at 50 °C for 15 min. Samples were pulsed in a centrifuge, and 150 μl of this solution was transferred into a glass vial and analyzed using ion exchange chromatography (DNAPac PA100 anion exchange column, Thermo Fisher) on a Shimadzu HPLC, with linear mobile phase gradient between buffer A (1:1 water:acetonitrile with 25 mM Tris-HCl and 1 mM EDTA, pH 8.5) and buffer B (buffer A plus 800 mM $NaClO_4$).

**Albumin recycling experiment.** A 12-well plate was collagen-coated and seeded with MDA-MB-231 WT or shFcRn cells. Cells were then incubated for 24 h with 500 nM Alexa Fluor 488-conjugated BSA (Thermo Fisher) in Opti-MEM, followed by a PBS wash and replacement with phenol red-free DMEM + 10% FBS. Media was harvested at 0, 1, 6, and 72 h after end of treatment. In total, 100 μl of media supernatant was transferred to a black 96-well plate (Corning) and fluorescence at excitation 488 nm/emission 530 nm was read on a Tecan Infinite M1000, with background fluorescence of media subtracted.

## FcRn interactions measured by NanoBRET

An established protocol[66,83] was used to leverage NanoBRET for measuring whether siRNA $< (EG_{18}L)_2$ binds to FcRn via an albumin bridge. To do so, human *FCGRT* in plasmid pFNK1 (Promega) was digested with Flexi Enzyme mix (Promega), resolved, and purified from 0.7% agarose. The *FCGRT* fragment was ligated into pFN31K-NanoLuc-CMV-Neo (Promega). Plasmids were prepared by miniprep (Qiagen) and used for transfection into HEK-293 cells (ATCC) using Fugene-HD transfection reagent (Promega) according to the manufacturer's instructions at an optimized 1:2 ratio (μg DNA:μl Fugene-HD) in 2.5% FBS. After 48 h, cells were treated with Opti-MEM (phenol-red free) supplemented with 700 nM bovine serum albumin (BSA, Sigma), 700 nM Alexa Fluor 647-conjugated BSA, and 1000 nM Cy5-conjugated si $< (EG_{18}L)_2$, where indicated. NanoBRET substrate ligand was added according to manufacturer's instruction at the indicated timepoint. Parallel samples were assessed in the absence of NanoBRET substrate. Emissions at 460 nm (NanoLuc emission) and at 670 nm (Cy5 emission) were measured on a Tecan Infinite M1000 as directed. The nanoBRET ratio was calculated as [(net emissions at 670 nm)/(net emissions at 460 nm) × 1000].

## Galectin-8 endosome disruption reporter assay

MDA-MB-231 cells expressing YFP-Gal8 were generated as previously published[68] and plated in a black 96-well plate 24 h prior to treatment with media only, lipofectamine RNAiMax with 50 nM "zipper" siRNA against luciferase, or 1 μM of $si_{Luc} < (EG_{18}L)_2$ in Opti-MEM with NucBlue live cell nuclear stain (Invitrogen). Cells were imaged at 18 h after initiation of treatment using a Nikon Ti confocal microscope.

## Lysotracker endo-lysosome colocalization imaging experiment

A black 96-well plate was coated with rat tail collagen I as above. MDA-MB-231 cells were plated at 5000 cells per well in DMEM + 10% FBS + pen/strep for 24 h. Cells were then treated with 100 nM Cy5-labeled $si_{Luc}$ or $si_{Luc} < (EG_{18}L)_2$ in Opti-MEM for (1) 3 h, (2) 24 h, or (3) 24 h followed by exchange with regular media for 24 h, $n = 3$ per condition and timepoint. After treatment, cells were incubated with full serum media with 75 nM Lysotracker DND-99 (Invitrogen) and NucBlue live cell nuclear dye (Invitrogen) for 1 h. Media was exchanged to Opti-MEM plus NucBlue live cell nuclear dye and imaged on a Nikon Ti confocal microscope. Images were exported using FIJI (ImageJ).

## Statistical analyses

Data were analyzed using GraphPad Prism 7 software (Graphpad Software, Inc.) Statistical tests used for each dataset are provided in the corresponding figure captions. For all figures, *$p \leq 0.05$, **$p \leq 0.01$, ***$p \leq 0.001$, ****$p \leq 0.0001$. All plots show mean ± standard deviation.

## Reporting summary

Further information on research design is available in the Nature Portfolio Reporting Summary linked to this article.

# Data availability

All data generated in this study are available within the article or Supplementary Information or upon reasonable request to the corresponding author.

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

## Acknowledgements

This work was supported by the National Institutes of Health (R01 CA224241 to C.L.D., R01 EB019409 to C.L.D., R01 CA260958 to C.L.D., R21 AR078636 to C.L.D., T32 CA217834, F32 CA268705, and K12 CA090625 to J.H.L.) and the National Science Foundation (BMAT 1349604 to C.L.D.). Flow cytometry experiments were partially performed in the VMC Flow Cytometry Shared Resource, supported by the Vanderbilt Ingram Cancer Center (P30 CA68485) and the Vanderbilt Digestive Disease Research Center (DK058404). We would additionally like to acknowledge the Translational Pathology Shared Resource supported by NCI/NIH Cancer Center Support grant 5P30 CA68485-19. All LC-MS characterization was performed in the Mass Spectrometry Research Center at Vanderbilt University. We also acknowledge Biorender.com for preparation of select schematics (Supplementary Figs. 16 and 17).

## Author contributions

E.N.H., C.L.D., and R.S.C. conceived and directed the project. E.N.H. performed all pharmacokinetic studies, binding studies, IVIS, and CMC studies. E.N.H. and S.S.P. oversaw and performed all flow cytometry on tumors. J.H.L. performed all in vitro mechanistic studies and assisted

with processing of in vivo samples. A.B.C., D.L.M., and R.M.A. performed size exclusion chromatography and subsequent analysis on murine plasma samples. E.N.H., L.M.B., E.F.G., and J.M. performed in vitro characterization of conjugates. E.N.H. and N.F. synthesized conjugates. P.P. synthesized and characterized lipid chemistry variants. J.C. performed IF analysis of tumor sections. V.M.S. performed IHC on tumor sections. J.T.M. and E.N.H. performed toxicity studies. L.H.L. screened siRNA sequences. J.C.R. aided in binding study assay development. L.D.S. performed EMSA studies using Bioanalyzer. J.L.H. assisted with processing of in vivo samples. K.C.V. provided input on experimental design and oversaw size exclusion chromatography. R.S.C. performed tumor measurement, ex vivo tumor analysis, Western blots, and in vitro Mcl-1 potency characterization. E.N.H. wrote the manuscript with editing by C.L.D., R.S.C., S.S.P., and J.H.L. All authors gave feedback and final approval.

## Competing interests

The authors declare no competing interests.
