## [Peer Review File · Nature Communications]

REVIEWER COMMENTS

Reviewer #1 - Albumin, carrier-free siRNA delivery (Remarks to the Author):

This work describes the optimisation of albumin-binding siRNA-lipid conjugates for gene silencing cancer therapeutics. The authors construct a library of siRNA-lipid conjugates varying in lipid structure and valency, branching, length and linker phosphorothioate (PS) content to investigate structure-function including albumin binding, pharmacokinetics, tumour accumulation and Luciferase, and MCL-1 gene silencing and anti-tumour effects in orthotopic murine models for triple negative breast cancer (TNBC). The paper is well written and has good flow. The work is interesting and relevant in the field of gene silencing therapeutics, however, a number of issues should be addressed.

1. The authors promote albumin-mediated delivery of modified siRNA. This has been previously been reported in the literature e.g. Bienk et al. (*Journal of Controlled Release*, 232, 2016), and other have utilised siRNA modifications for binding to endogenous proteins such as lipoproteins, e.g. Soutschek (*Nature*, 432, 2004) and Wolfrum et al. (*Nature Biotechnology*, 25, 2007). These studies did not focus on cancer therapy but the concept of siRNA binding to endogenous proteins introduced. The novelty of the concept and work in this manuscript should be made clear.
2. What is the route of cellular entry and intracellular trafficking of the lipid-siRNA that enables engagement with the mRNA target in the cytosol? Is it still associated with albumin and susceptible to FcRn-driven endosomal recycling and subsequent release from the cell?, or is it internalised with albumin during albumin catabolism. Or is it released from albumin prior to entering independently by hydrophobic interactions, if so, the possible role of fatty acid cellular receptors should be mentioned. On page 14, the authors state “This result led us to postulate that albumin-bound siRNAs in general and si<(EG18L)diacid (either bound or free form) enter the cell by a form of active endocytosis, while si<(EG18L)2 does not”. The mechanism for siRNA trafficking needs to be more clearly explained, and the mechanism by which siRNA escapes the endosome required for gene silencing discussed.
3. It would be relevant to compare the “zipper pattern” siRNA in vitro silencing to conventional siRNA of similar length (without 2'-O-M modifications) in addition to the longer Dicer siRNA including 2'-O-Me modifications (Supplementary Figure 2). It would be interesting if the 2'-O-Me modifications in the “zipper pattern” format confers inactivation of Toll-Like Receptor induction of innate immunity as Dicer siRNA 2'-O-Me do.
4. Biodistribution and circulatory half-life was evaluated with fluorescently labelled siRNA-lipid conjugate (Supplementary Figure 3, 5, and Figure 2, Figure 4, Figure 5, Extended Data Figure 1, 3 and 4). It would be relevant to include albumin binding and possible micelle formation data (similar to shown in Figure 1) on fluorescently labeled siRNA-lipids before in vivo work to determine if labelling affected the albumin binding characteristics of the siRNA. Is the BLI albumin-binding data in Figure 5c for fluorescent siRNA conjugate?

It would be relevant to include a comment in the discussion on the correlation between binding and activity e.g. the level of albumin-binding required in order to allow siRNA delivery but also allow siRNA release for gene silencing.

5. The circulatory half-life of the designs are shown in supplementary Table 2, with 64 mins for si<(EG18L)2. The authors should mention in the text, data reported in the literature for alternative albumin-binding siRNA-conjugates.

In Figure 2b it states si<(EG36L2), should it be si<(EG30L)?

6. Was a siRNA-lipid scrambled sequence used in the in vivo luciferase silencing work in Figure 3c? This is relevant to determine any potential influence of the modification on gene silencing. It is unclear if a scrambled control for the MCL-1 silencing work in Figure 3f was used, the caption states “all values measured in non-targeting controls” that suggests a scrambled sequence control was used. This should be clarified.

It is relevant to include in vitro silencing data for the panel of siRNA-lipid conjugates used for the in vivo work in Figure 3. This is relevant to understand why formulation may show differences in vivo silencing (Figure 3c).

7. Can the authors comment on the modest in vitro gene silencing observed in Supplementary Figure 15a.

8. Could the increased accumulation of the divalent lipid-siRNA conjugate in the lungs (Supplementary Figure 3) reflect a degree of serum-induced aggregation of the more hydrophobic conjugate compared to a monovalent lipid conjugate?

9. In the conclusion, the authors state “Overall, this work has important implications for the delivery of siRNAs to extrahepatic targets, a goal that has remained clinically elusive”. There are numerous reports in the literature of siRNA-based effects in extrahepatic tumour models. The authors should refine this statement.

10. MCL-1 or MCL1 should be written consistently in the manuscript as write both

Reviewer #2 - Breast cancer therapy (Remarks to the Author):

In this paper entitled “structural optimization of siRNA conjugates for albumin binding achieves effective MCL1-directed cancer therapy”, the author presented a systemic analysis of lipid library modified siRNA for enhanced binding to albumin as an efficient method for siRNA delivery platform. The author synthesized a series of various lipid combination as an optimization technique for obtaining the best siRNA-binding albumin platform. Although albumin-binding siRNA concept is not new, the author did bring to light the underlying mechanism of lipid-albumin binding mechanism that could be useful for future development of albumin-binding platform. Although the paper is logical and the author

presented ample research results to prove their hypothesis, there are some crucial problems that need to be addressed prior to the consideration of acceptance of this paper.

Major comments:

1. The author strongly suggests that the albumin-binding of the lipid-modified siRNA could work better than siRNA-albumin direct link platform. If this is true, then, the lipid-linked siRNA (especially their optimized si<(EG18L)2) should show a preferentially higher albumin binding after systemic injection. The author should provide evidence that si<(EG18L)2 also works well after systemic injection (without prior albumin binding). This could be another significant result that could be further translated into clinical application.

2. The author mentioned that “the in vivo half-life of human albumin is an exceptional 19 days”. However, there is no indication that the author’s si<(EG18L)2 system could be stable for siRNA (for 19 days) even after modification. Therefore, it is possible that in the systemic circulation, part of the si<(EG18L)2 could be dissociated prior to the 19 days, with reduced efficacy. What is the half-life of si<(EG18L)2 after dissociation from albumin in the circulation?

3. Some of the comments made by the author are not supported by solid evidence. One of such examples is when the author claimed that “while biodistribution to other organ is similar, siRNA-L2 was modestly increased in lungs, a highly vascularized tissues, perhaps reflecting circulating conjugate at this time point”, and the author cited Figure S3). However, Fig S3 did not show the relationship between highly vascularized tissues with the amount of conjugate, whether or not it is conjugated. Please justify this claim.

4. A similar problem is also found in the sentence of “... demonstrating progressively increasing $t_{1/2}$ with increasing EG repeats.... Followed by ... suggesting a limit to the positive impact of increasing linker length on circulation half-life”. The author used EG18 followed by EG36 (or EG30, there are two different EGs given in Figure 2B-E, unsure if this is an error). There is no systemic analysis in between these two lengths (i.e. EG20, EG25, EG30), therefore, the statement is not entirely accurate.

5. In the sentence “to access the knockdown of the endogenous oncogenic driver MCL-1 in tumors...” the author mentioned that “...revealing dose-dependent MCL-1 knockdown at day 4 and day 8 (Fig. 3F). this sentence indicated two confusions: (i) the treatment was done twice at day 4 and day 8, or (ii) the knockdown only shows dose-dependency at day 4 and day 8, which were not true on both contexts. Please reword this sentence.

6. In the subsection “albumin-binding increases carrier free tumor accumulation...”, the author is emphasizing on the point that the delivered si(EG18L)2 is indeed dissociated from albumin in order for the siRNA to exert its effect. Therefore, exact measurement of dissociation would be crucial to prove this point, including whether or not the dissociation only occurs near/at the tumor?

7. Although the binding (association) to albumin is important, the dissociation of siRNA from albumin is also important, in order for the siRNA to be uptaken into the cell cytoplasm. At what condition will these si(EG18L)2 dissociate from albumin? What is the underlying mechanism of the dissociation? How can dissociated si(EG18L)2 effectively enter cell cytoplasm without the help of targeting ligand or cell penetrating motif?

8. In figure 3F, please provide all concentrations for both treatment days. Current figure for Day 8 only has 10 and 20 mg/kg treatment data.

9. In Figure 5, the in vivo analysis should be done in all three lipid variations. Current data only showed two lipid variation done on in vivo analysis. Furthermore, the si(EG18Lunsat)2 albumin-bound ability being lesser than the other two is not fully convincing. The data presented have large variations inside the groups.

Minor comments:

1. Please provide the country/region where the companies are based (in Materials and Methods).

2. Please be careful with the labeling of figures. All figure should have their own labeling. For example, Figure 5 is missing two labels (one on the right of figure 5c and another one the right of figure 5e).

REVIEWER COMMENTS

Reviewer #1 - Albumin, carrier-free siRNA delivery (Remarks to the Author):

This work describes the optimisation of albumin-binding siRNA-lipid conjugates for gene silencing cancer therapeutics. The authors construct a library of siRNA-lipid conjugates varying in lipid structure and valency, branching, length and linker phosphorothioate (PS) content to investigate structure-function including albumin binding, pharmacokinetics, tumour accumulation and Luciferase, and MCL-1 gene silencing and anti-tumour effects in orthotopic murine models for triple negative breast cancer (TNBC). The paper is well written and has good flow. The work is interesting and relevant in the field of gene silencing therapeutics, however, a number of issues should be addressed.

1. The authors promote albumin-mediated delivery of modified siRNA. This has been previously been reported in the literature e.g. Bienk et al. (Journal of Controlled Release, 232, 2016), and other have utilised siRNA modifications for binding to endogenous proteins such as lipoproteins, e.g. Soutschek (Nature, 432, 2004) and Wolfrum et al. (Nature Biotechnology, 25, 2007). These studies did not focus on cancer therapy but the concept of siRNA binding to endogenous proteins introduced. The novelty of the concept and work in this manuscript should be made clear.

We thank the reviewer for providing additional literature to contextualize our work. We have amended the manuscript to include references to these works in our introduction where we introduce the concept of binding to plasma components using modifications to siRNA. We also accentuate the novelty of our chemistry and application to cancer. Though it would probably be a distraction to extensively describe each of these papers in detail in the introduction, we would like to clarify here that:

-Bienk et al pre-formulate cholesterol conjugates with HSA prior to treatment as a solubilization strategy and look for liver silencing

-Soutschek et al use a cholesterol conjugate and look for liver silencing

-Wolfrum et al screen different conjugates but do not attempt to optimize a system around albumin binding. They do pre-complexation with albumin vs lipoproteins with various conjugates and measure systemic biodistribution of these pre-formulated systems. There is no in vivo silencing explored.

Our data and others suggest that cholesterol conjugates preferentially associate with lipoproteins and distribute to liver in vivo. None of these studies took on a structural optimization to discover and optimize structures for albumin association, nor for albumin association in situ (i.e, does not have to be pre-complexed). None of these studies looked at linker features for lipophilic conjugates. None of them explore an extrahepatic application of any type (much less tumor delivery). We contend that our work is a very significant contribution that is novel compared to these previous, seminal contributions.

2. What is the route of cellular entry and intracellular trafficking of the lipid-siRNA that enables engagement with the mRNA target in the cytosol? Is it still associated with albumin and

susceptible to FcRn-driven endosomal recycling and subsequent release from the cell?, or is it internalised with albumin during albumin catabolism. Or is it released from albumin prior to entering independently by hydrophobic interactions, if so, the possible role of fatty acid cellular receptors should be mentioned. On page 14, the authors state “This result led us to postulate that albumin-bound siRNAs in general and si<(EG18L)diacid) (either bound or free form) enter the cell by a form of active endocytosis, while si<(EG18L)2 does not”. The mechanism for siRNA trafficking needs to be more clearly explained, and the mechanism by which siRNA escapes the endosome required for gene silencing discussed.

In **Extended Data Fig. 4A-B**, we demonstrate that rate of cell uptake of our optimized si<(EG₁₈L)₂ is higher in the free conjugate than the albumin-bound form. Furthermore, the free conjugates robustly associate with cells in the absence of albumin in an energy-independent manner, whereas internalization of the albumin-bound form is endocytosis dependent. Furthermore, comparison to the diacid conjugate reveals that the hydrophobicity of the lipids in the si<(EG₁₈L)₂ conjugate is integral leveraging this more rapid, endocytosis-independent uptake pathway.

We also sought to interrogate whether FcRn recycling plays a role in intracellular accumulation of our conjugates. We first sought to measure if there is an interaction between si<(EG₁₈L)₂ and FcRn via albumin bridging by generating a bioluminescence resonance energy transfer (BRET) reporter cell line with NanoLuciferase fused to FcRn (**Supplementary Fig. S16**). In this assay, proximity between our fluorophore-labeled binder of interest and our NanoLuciferase-tagged FcRn should generate a quantifiable BRET signal. We validate this with fluorophore-labeled BSA and demonstrated that an interaction does occur between si<(EG₁₈L)₂ in the presence but not the absence of albumin (**Extended Data Fig. 4C**).

To further probe this mechanism, we generated MDA-MB-231 cells with FcRn knocked out using shRNA lentiviral particles (**Extended Data Fig. 4D**). We validated knockdown of this protein by Western blot and validated that our FcRn knockout line shows lower albumin recycling than its wild-type counterpart (**Supplementary Fig. S17**). We used this new assay to demonstrate that there is a statistically significant but numerically small decrease in conjugate recycling out of the cell in the presence of albumin between shFcRn cells and their parent cells (**Extended Data Fig. 4E**). However, in the absence of albumin, there was no difference in recycling between these two cell lines (**Extended Data Fig. 4F**). These collective data suggest that si<(EG₁₈L)₂ does bind to FcRn via an albumin bridge, and that delivery via a non-albumin bound mechanism may improve retention due to use of an alternative mechanism of uptake and due to the resistance of the free conjugate to potential FcRn-mediated recycling back out of the cell.

To the query on the mechanism of endosomal escape, we used a Galectin-8 (Gal-8) reporter assay in MDA-MB-231 cells that can be used to visualize endosomal disruption (**Supplementary Fig. S18**). We demonstrate that, in contrast to cationic transfection agents, si<(EG₁₈L)₂ does not produce Gal-8 foci indicative of endosomal disruption. Yet, after 24h of incubation with si<(EG₁₈L)₂ (**Supplementary Fig. S18D**), we observe cytoplasmic distribution of the conjugate, consistent with our data confirming conjugate silencing ability. We include discussion in our manuscript on the possibility of our conjugates facilitating fusion with endosomal membranes without causing membrane disruption.

3. It would be relevant to compare the “zipper pattern” siRNA in vitro silencing to conventional siRNA of similar length (without 2’O-M modifications) in addition to the longer Dicer siRNA including 2’-O-Me modifications (Supplementary Figure 2). It would be interesting if the 2’-O-Me modifications in the “zipper pattern” format confers inactivation of Toll-Like Receptor induction of innate immunity as Dicer siRNA 2’-O-Me do.

It is well established that 2’OMe modifications block the potential of siRNAs to nonspecifically activate TLR pathways (Dalpke (2012). *RNA biology*, 9(6), 828-842). Given that our constructs are fully modified at the 2’OH position, the zipper pattern does not risk TLR activation.

4. Biodistribution and circulatory half-life was evaluated with fluorescently labelled siRNA-lipid conjugate (Supplementary Figure 3, 5, and Figure 2, Figure 4, Figure 5, Extended Data Figure 1, 3 and 4). It would be relevant to include albumin binding and possible micelle formation data (similar to shown in Figure 1) on fluorescently labeled siRNA-lipids before in vivo work to determine if labelling affected the albumin binding characteristics of the siRNA. Is the BLI albumin-binding data in Figure 5c for fluorescent siRNA conjugate?

As a point of clarification, we assessed the binding of both fluorescently labeled and unlabeled conjugates. The biolayer interferometry (BLI) measurements were done on conjugates that did not have a fluorophore label. By comparison, binding to serum proteins measured by SEC utilizes a fluorophore labeled duplex. The position of the fluorophore is kept the same throughout the conjugate variants, minimizing any skewed interpretation within a given experiment due to effects of the fluorophore. Importantly, the fluorophore used in our studies is located on the opposite end of the siRNA duplex (5’ antisense) as the lipophilic moiety (5’ sense). This site is, therefore, not anticipated to participate in lipid-driven micellization. We have also clarified this in the manuscript **Materials and Methods section, Conjugate Synthesis, Purification, and Validation**.

It would be relevant to include a comment in the discussion on the correlation between binding and activity e.g. the level of albumin-binding required in order to allow siRNA delivery but also allow siRNA release for gene silencing.

In the section of our manuscript titled “**Hydrophobicity of lipid and not just high albumin binding affinity is important for conjugate performance**,” we discuss the desirability of reversible albumin binding and lipid hydrophobicity for maximizing cell uptake. We also provide new data on the role of FcRn in recycling of siRNA conjugates bound vs unbound to albumin in **Extended Data Fig. 4**.

5. The circulatory half-life of the designs are shown in supplementary Table 2, with 64 mins for si<(EG18L)2. The authors should mention in the text, data reported in the literature for alternative albumin-binding siRNA-conjugates.

While we agree with the reviewer that it is important to contextualize the half-life findings here, it is difficult to draw parallels to alternative albumin-binding siRNA conjugates in the literature. Most of the reported constructs are based on siRNAs that are not metabolically stabilized,

greatly limiting the maximum half-life that they can achieve. To our knowledge, there is not any other half-life reported for metabolically stabilized siRNA duplexes that bind to albumin.

In Figure 2b it states si<(EG36L2), should it be si<(EG30L)?

Thank you for catching this typo- this error has been fixed.

6. Was a siRNA-lipid scrambled sequence used in the in vivo luciferase silencing work in Figure 3c? This is relevant to determine any potential influence of the modification on gene silencing. It is unclear if a scrambled control for the MCL-1 silencing work in Figure 3f was used, the caption states “all values measured in non-targeting controls” that suggests a scrambled sequence control was used. This should be clarified.

We thank the reviewer for pointing this out. In this figure set, the values were normalized to the saline-treated control mice which we have now clarified in the caption. However, in our studies targeting MCL-1, including all of the therapeutic studies, we compared to a negative control siRNA sequence.

It is relevant to include in vitro silencing data for the panel of siRNA-lipid conjugates used for the in vivo work in Figure 3. This is relevant to understand why formulation may show differences in vivo silencing (Figure 3c).

We have now added in vitro silencing data for the series of siRNA-lipid conjugates from Figure 3 in **Supplementary Fig. S15** and described our findings under the section labeled “**Mechanisms of si<(EG₁₈L)₂ cell internalization and trafficking.**”

7. Can the authors comment on the modest in vitro gene silencing observed in Supplementary Figure 15a.

The in vitro gene silencing shown in Supplementary Figure 15A (now **Supplementary Fig. S19A**) is based on carrier-free silencing by the siRNA conjugate. In the absence of a transfection reagent or endosome escaping reagent, this silencing should be considered quite robust. Generally, silencing can be very robust over longer-term in vivo studies with carrier free siRNAs that are chemically modified for stability in the endo-lysosomes. However, the knockdown effect is less acutely potent and slower to peak as the siRNAs slowly leak out of the endo-lysosomal “depots”. As a result, carrier free siRNA has reduced potency vs transfection agent delivery in short-term in vitro studies. Importantly, carrier free conjugates can be just as potent and often longer-lasting in vivo in the absence of an endosome disrupting agent. Please see: doi: [10.1093/nar/gkaa670](https://doi.org/10.1093/nar/gkaa670).

8. Could the increased accumulation of the divalent lipid-siRNA conjugate in the lungs (Supplementary Figure 3) reflect a degree of serum-induced aggregation of the more hydrophobic conjugate compared to a monovalent lipid conjugate?

We do not believe that is the case. If the hydrophobic conjugates were aggregating, we would be able to visualize that in the vasculature in our intravital microscopy measurements. We would

also expect to be able to capture such a phenomenon with our SEC experiments. We have now added the SEC traces comparing the monovalent and divalent conjugates in **Supplementary Fig. S3E** to demonstrate that the binding profiles are comparable, indicating that divalent conjugate aggregation does not account for the difference in biodistribution.

9. In the conclusion, the authors state “Overall, this work has important implications for the delivery of siRNAs to extrahepatic targets, a goal that has remained clinically elusive”. There are numerous reports in the literature of siRNA-based effects in extrahepatic tumour models. The authors should refine this statement.

We would like to emphasize the use of the word “clinically” in our statement. We intended for this statement to acknowledge that siRNA delivery to extrahepatic sites (e.g., tumors) has only achieved preclinical, rather than clinical, success. We aim to convey that our work herein adds to the body of knowledge that seeks to bridge this gap.

10. MCL-1 or MCL1 should be written consistently in the manuscript as write both
We have reviewed the text to ensure that when we are describing the gene, we have used *MCL1*, and when we are referring to the protein, we have used MCL-1.

Reviewer #2 - Breast cancer therapy (Remarks to the Author):

In this paper entitled “structural optimization of siRNA conjugates for albumin binding achieves effective MCL1-directed cancer therapy”, the author presented a systemic analysis of lipid library modified siRNA for enhanced binding to albumin as an efficient method for siRNA delivery platform. The author synthesized a series of various lipid combination as an optimization technique for obtaining the best siRNA-binding albumin platform. Although albumin-binding siRNA concept is not new, the author did bring to light the underlying mechanism of lipid-albumin binding mechanism that could be useful for future development of albumin-binding platform. Although the paper is logical and the author presented ample research results to prove their hypothesis, there are some crucial problems that need to be addressed prior to the consideration of acceptance of this paper.

Major comments:

1. The author strongly suggests that the albumin-binding of the lipid-modified siRNA could work better than siRNA-albumin direct link platform. If this is true, then, the lipid-linked siRNA (especially their optimized si<(EG18L)2) should show a preferentially higher albumin binding after systemic injection. The author should provide evidence that si<(EG18L)2 also works well after systemic injection (without prior albumin binding). This could be another significant result that could be further translated into clinical application.

As a point of clarification, si<(EG₁₈L)₂, and the other conjugates interrogated in this work are not pre-incubated with albumin prior to injection. Rather, they associate in situ with endogenous albumin. We apologize for the confusion and have clarified this in the introduction with the statement that we enabled “development of siRNAs that are intravenously injected carrier-free and can dock onto serum albumin as an endogenous carrier”.

2. The author mentioned that “the in vivo half-life of human albumin is an exceptional 19 days”. However, there is no indication that the author’s si<(EG18L)₂ system could be stable for siRNA (for 19 days) even after modification. Therefore, it is possible that in the systemic circulation, part of the si<(EG18L)₂ could be dissociated prior to the 19 days, with reduced efficacy. What is the half-life of si<(EG18L)₂ after dissociation from albumin in the circulation?

Based on the above comment, we believe that the reviewer is asking this question with the notion that our conjugate is premixed with exogenous albumin. However, this is not the case. We do show by SEC that the majority of the si<(EG₁₈L)₂ conjugate is albumin bound in the circulation after i.v. delivery. As an additional comment, there is a huge excess of albumin in the blood relative to the conjugate concentration, and we do anticipate that binding to endogenous albumin is a dynamic process wherein a single conjugate molecule can hop on and off of multiple albumin molecules during transit. It is not possible, ultimately, for us to decouple the circulation half-life of the bound vs. unbound structure. However, generally, non-conjugate siRNAs are rapidly filtered out through the kidneys in ~5 mins, and we anticipate the conjugate would suffer the same fate if not bound to any type of carrier protein / serum component.

3. Some of the comments made by the author are not supported by solid evidence. One of such examples is when the author claimed that “while biodistribution to other organ is similar, siRNA-L2 was modestly increased in lungs, a highly vascularized tissues, perhaps reflecting circulating conjugate at this time point”, and the author cited Figure S3). However, Fig S3 did not show the relationship between highly vascularized tissues with the amount of conjugate, whether or not it is conjugated. Please justify this claim.

We agree about the speculative nature of this comment and have removed it from the manuscript.

4. A similar problem is also found in the sentence of “... demonstrating progressively increasing t_{1/2} with increasing EG repeats.... Followed by ... suggesting a limit to the positive impact of increasing linker length on circulation half-life”. The author used EG18 followed by EG36 (or EG30, there are two different EGs given in Figure 2B-E, unsure if this is an error). There is no systemic analysis in between these two lengths (i.e. EG20, EG25, EG30), therefore, the statement is not entirely accurate.

We have amended this sentence to remove the suggestion that we have progressively and systematically increased the EG units. We now state more broadly that we observed a general increase in half-life with increased EG units.

5. In the sentence “to access the knockdown of the endogenous oncogenic driver MCL-1 in tumors...” the author mentioned that “...revealing dose-dependent MCL-1 knockdown at day 4 and day 8 (Fig. 3F). this sentence indicated two confusions: (i) the treatment was done twice at day 4 and day 8, or (ii) the knockdown only shows dose-dependency at day 4 and day 8, which were not true on both contexts. Please reword this sentence.

We have amended the text to read, “A single bolus was delivered intravenously to MDA-MB-231 tumor bearing mice (Fig. 3E), revealing dose-dependent *MCL1* knockdown in tumors harvested at either 4 or 8 days after treatment (Fig. 3F), with as much as 85% *MCL1* knockdown.”

6. In the subsection “albumin-binding increases carrier free tumor accumulation...”, the author is emphasizing on the point that the delivered si<(EG18L)2 is indeed dissociated from albumin in order for the siRNA to exert its effect. Therefore, exact measurement of dissociation would be crucial to prove this point, including whether or not the dissociation only occurs near/at the tumor?

We have modified this statement- the intention was to communicate a carrier in the traditional sense such as a nanoparticle. We understand that a reader could interpret the use of “carrier” here to refer to albumin. We have now changed the wording to “**Albumin-binding increases tumor accumulation of siRNA-lipid conjugates and tumor gene silencing.**” We would also like to emphasize that we did measure albumin association and dissociation constants by BLI and also quantify the relative amount of bound/unbound conjugate in plasma from i.v.-treated animals using SEC.

7. Although the binding (association) to albumin is important, the dissociation of siRNA from albumin is also important, in order for the siRNA to be uptaken into the cell cytoplasm. **At what condition will these si<(EG18L)2 dissociate from albumin? What is the underlying mechanism of the dissociation?** How can dissociated si<(EG18L)2 effectively enter cell cytoplasm without the help of targeting ligand or cell penetrating motif?

Albumin has natural fatty acid pockets that interact through non-covalent binding (Van der Vusse, *Drug metabolism and pharmacokinetics*, 24(4),300-307). To this end, there is a natural on rate and off rate involved in this type of binding. We demonstrated that dissociated si<(EG18L)₂ enters the cell more effectively in the absence of albumin, though can also enter in albumin bound form (**Extended Data Figure 4**). The unbound conjugate can associate with cells even at 4 C, wherein endocytosis is inhibited. These results suggest that the lipid tails are able to bind to and potentially contribute to traversal of the cellular membrane to promote entry.

8. In figure 3F, please provide all concentrations for both treatment days. Current figure for Day 8 only has 10 and 20 mg/kg treatment data.

We only tested the two higher concentrations from day 4 to determine whether silencing was sustained at this dose level until day 8 for the sake of conserving animals.

9. In Figure 5, the in vivo analysis should be done in all three lipid variations. Current data only showed two lipid variation done on in vivo analysis. Furthermore, the si<(EG18Lunsat)2 albumin-bound ability being lesser than the other two is not fully convincing. The data presented have large variations inside the groups.

None of our experiments to survey the performance of this structure showed any indication that the unsaturated conjugate may be superior to the saturated one. We, therefore, determined that further analysis was outside of the scope of these studies.

Minor comments:

1. Please provide the country/region where the companies are based (in Materials and Methods).

We have added these details.

2. Please be careful with the labeling of figures. All figure should have their own labeling. For example, Figure 5 is missing two labels (one on the right of figure 5c and another one the right of figure 5e).

We have now added individual labels to the panel in Figure 5 and made this consistent in the manuscript.

REVIEWERS' COMMENTS

Reviewer #1 (Remarks to the Author):

The authors have adequately addressed the comments of the reviewer with modifications to the text and additional experiments.

Reviewer #2 (Remarks to the Author):

The author has adequately addressed all the reviewer's concerns and should be considered to be published in its current form.

REVIEWER COMMENTS

Reviewer #1 - Albumin, carrier-free siRNA delivery (Remarks to the Author):

This work describes the optimisation of albumin-binding siRNA-lipid conjugates for gene silencing cancer therapeutics. The authors construct a library of siRNA-lipid conjugates varying in lipid structure and valency, branching, length and linker phosphorothioate (PS) content to investigate structure-function including albumin binding, pharmacokinetics, tumour accumulation and Luciferase, and MCL-1 gene silencing and anti-tumour effects in orthotopic murine models for triple negative breast cancer (TNBC). The paper is well written and has good flow. The work is interesting and relevant in the field of gene silencing therapeutics, however, a number of issues should be addressed.

1. The authors promote albumin-mediated delivery of modified siRNA. This has been previously been reported in the literature e.g. Bienk et al. (Journal of Controlled Release, 232, 2016), and other have utilised siRNA modifications for binding to endogenous proteins such as lipoproteins, e.g. Soutschek (Nature, 432, 2004) and Wolfrum et al. (Nature Biotechnology, 25, 2007). These studies did not focus on cancer therapy but the concept of siRNA binding to endogenous proteins introduced. The novelty of the concept and work in this manuscript should be made clear.

We thank the reviewer for providing additional literature to contextualize our work. We have amended the manuscript to include references to these works in our introduction where we introduce the concept of binding to plasma components using modifications to siRNA. We also accentuate the novelty of our chemistry and application to cancer. Though it would probably be a distraction to extensively describe each of these papers in detail in the introduction, we would like to clarify here that:

-Bienk et al pre-formulate cholesterol conjugates with HSA prior to treatment as a solubilization strategy and look for liver silencing

-Soutschek et al use a cholesterol conjugate and look for liver silencing

-Wolfrum et al screen different conjugates but do not attempt to optimize a system around albumin binding. They do pre-complexation with albumin vs lipoproteins with various conjugates and measure systemic biodistribution of these pre-formulated systems. There is no in vivo silencing explored.

Our data and others suggest that cholesterol conjugates preferentially associate with lipoproteins and distribute to liver in vivo. None of these studies took on a structural optimization to discover and optimize structures for albumin association, nor for albumin association in situ (i.e, does not have to be pre-complexed). None of these studies looked at linker features for lipophilic conjugates. None of them explore an extrahepatic application of any type (much less tumor delivery). We contend that our work is a very significant contribution that is novel compared to these previous, seminal contributions.

2. What is the route of cellular entry and intracellular trafficking of the lipid-siRNA that enables engagement with the mRNA target in the cytosol? Is it still associated with albumin and

susceptible to FcRn-driven endosomal recycling and subsequent release from the cell?, or is it internalised with albumin during albumin catabolism. Or is it released from albumin prior to entering independently by hydrophobic interactions, if so, the possible role of fatty acid cellular receptors should be mentioned. On page 14, the authors state “This result led us to postulate that albumin-bound siRNAs in general and si<(EG18L)diacid) (either bound or free form) enter the cell by a form of active endocytosis, while si<(EG18L)₂ does not”. The mechanism for siRNA trafficking needs to be more clearly explained, and the mechanism by which siRNA escapes the endosome required for gene silencing discussed.

In **Extended Data Fig. 4A-B**, we demonstrate that rate of cell uptake of our optimized si<(EG₁₈L)₂ is higher in the free conjugate than the albumin-bound form. Furthermore, the free conjugates robustly associate with cells in the absence of albumin in an energy-independent manner, whereas internalization of the albumin-bound form is endocytosis dependent. Furthermore, comparison to the diacid conjugate reveals that the hydrophobicity of the lipids in the si<(EG₁₈L)₂ conjugate is integral leveraging this more rapid, endocytosis-independent uptake pathway.

We also sought to interrogate whether FcRn recycling plays a role in intracellular accumulation of our conjugates. We first sought to measure if there is an interaction between si<(EG₁₈L)₂ and FcRn via albumin bridging by generating a bioluminescence resonance energy transfer (BRET) reporter cell line with NanoLuciferase fused to FcRn (**Supplementary Fig. S16**). In this assay, proximity between our fluorophore-labeled binder of interest and our NanoLuciferase-tagged FcRn should generate a quantifiable BRET signal. We validate this with fluorophore-labeled BSA and demonstrated that an interaction does occur between si<(EG₁₈L)₂ in the presence but not the absence of albumin (**Extended Data Fig. 4C**).

To further probe this mechanism, we generated MDA-MB-231 cells with FcRn knocked out using shRNA lentiviral particles (**Extended Data Fig. 4D**). We validated knockdown of this protein by Western blot and validated that our FcRn knockout line shows lower albumin recycling than its wild-type counterpart (**Supplementary Fig. S17**). We used this new assay to demonstrate that there is a statistically significant but numerically small decrease in conjugate recycling out of the cell in the presence of albumin between shFcRn cells and their parent cells (**Extended Data Fig. 4E**). However, in the absence of albumin, there was no difference in recycling between these two cell lines (**Extended Data Fig. 4F**). These collective data suggest that si<(EG₁₈L)₂ does bind to FcRn via an albumin bridge, and that delivery via a non-albumin bound mechanism may improve retention due to use of an alternative mechanism of uptake and due to the resistance of the free conjugate to potential FcRn-mediated recycling back out of the cell.

To the query on the mechanism of endosomal escape, we used a Galectin-8 (Gal-8) reporter assay in MDA-MB-231 cells that can be used to visualize endosomal disruption (**Supplementary Fig. S18**). We demonstrate that, in contrast to cationic transfection agents, si<(EG₁₈L)₂ does not produce Gal-8 foci indicative of endosomal disruption. Yet, after 24h of incubation with si<(EG₁₈L)₂ (**Supplementary Fig. S18D**), we observe cytoplasmic distribution of the conjugate, consistent with our data confirming conjugate silencing ability. We include discussion in our manuscript on the possibility of our conjugates facilitating fusion with endosomal membranes without causing membrane disruption.

3. It would be relevant to compare the “zipper pattern” siRNA in vitro silencing to conventional siRNA of similar length (without 2’O-M modifications) in addition to the longer Dicer siRNA including 2’-O-Me modifications (Supplementary Figure 2). It would be interesting if the 2’-O-Me modifications in the “zipper pattern” format confers inactivation of Toll-Like Receptor induction of innate immunity as Dicer siRNA 2’-O-Me do.

It is well established that 2’OMe modifications block the potential of siRNAs to nonspecifically activate TLR pathways (Dalpke (2012). *RNA biology*, 9(6), 828-842). Given that our constructs are fully modified at the 2’OH position, the zipper pattern does not risk TLR activation.

4. Biodistribution and circulatory half-life was evaluated with fluorescently labelled siRNA-lipid conjugate (Supplementary Figure 3, 5, and Figure 2, Figure 4, Figure 5, Extended Data Figure 1, 3 and 4). It would be relevant to include albumin binding and possible micelle formation data (similar to shown in Figure 1) on fluorescently labeled siRNA-lipids before in vivo work to determine if labelling affected the albumin binding characteristics of the siRNA. Is the BLI albumin-binding data in Figure 5c for fluorescent siRNA conjugate?

As a point of clarification, we assessed the binding of both fluorescently labeled and unlabeled conjugates. The biolayer interferometry (BLI) measurements were done on conjugates that did not have a fluorophore label. By comparison, binding to serum proteins measured by SEC utilizes a fluorophore labeled duplex. The position of the fluorophore is kept the same throughout the conjugate variants, minimizing any skewed interpretation within a given experiment due to effects of the fluorophore. Importantly, the fluorophore used in our studies is located on the opposite end of the siRNA duplex (5’ antisense) as the lipophilic moiety (5’ sense). This site is, therefore, not anticipated to participate in lipid-driven micellization. We have also clarified this in the manuscript **Materials and Methods section, Conjugate Synthesis, Purification, and Validation**.

It would be relevant to include a comment in the discussion on the correlation between binding and activity e.g. the level of albumin-binding required in order to allow siRNA delivery but also allow siRNA release for gene silencing.

In the section of our manuscript titled “**Hydrophobicity of lipid and not just high albumin binding affinity is important for conjugate performance**,” we discuss the desirability of reversible albumin binding and lipid hydrophobicity for maximizing cell uptake. We also provide new data on the role of FcRn in recycling of siRNA conjugates bound vs unbound to albumin in **Extended Data Fig. 4**.

5. The circulatory half-life of the designs are shown in supplementary Table 2, with 64 mins for si<(EG18L)2. The authors should mention in the text, data reported in the literature for alternative albumin-binding siRNA-conjugates.

While we agree with the reviewer that it is important to contextualize the half-life findings here, it is difficult to draw parallels to alternative albumin-binding siRNA conjugates in the literature. Most of the reported constructs are based on siRNAs that are not metabolically stabilized,

greatly limiting the maximum half-life that they can achieve. To our knowledge, there is not any other half-life reported for metabolically stabilized siRNA duplexes that bind to albumin.

In Figure 2b it states si<(EG36L2), should it be si<(EG30L)?

Thank you for catching this typo- this error has been fixed.

6. Was a siRNA-lipid scrambled sequence used in the in vivo luciferase silencing work in Figure 3c? This is relevant to determine any potential influence of the modification on gene silencing. It is unclear if a scrambled control for the MCL-1 silencing work in Figure 3f was used, the caption states “all values measured in non-targeting controls” that suggests a scrambled sequence control was used. This should be clarified.

We thank the reviewer for pointing this out. In this figure set, the values were normalized to the saline-treated control mice which we have now clarified in the caption. However, in our studies targeting MCL-1, including all of the therapeutic studies, we compared to a negative control siRNA sequence.

It is relevant to include in vitro silencing data for the panel of siRNA-lipid conjugates used for the in vivo work in Figure 3. This is relevant to understand why formulation may show differences in vivo silencing (Figure 3c).

We have now added in vitro silencing data for the series of siRNA-lipid conjugates from Figure 3 in **Supplementary Fig. S15** and described our findings under the section labeled “**Mechanisms of si<(EG₁₈L)₂ cell internalization and trafficking.**”

7. Can the authors comment on the modest in vitro gene silencing observed in Supplementary Figure 15a.

The in vitro gene silencing shown in Supplementary Figure 15A (now **Supplementary Fig. S19A**) is based on carrier-free silencing by the siRNA conjugate. In the absence of a transfection reagent or endosome escaping reagent, this silencing should be considered quite robust. Generally, silencing can be very robust over longer-term in vivo studies with carrier free siRNAs that are chemically modified for stability in the endo-lysosomes. However, the knockdown effect is less acutely potent and slower to peak as the siRNAs slowly leak out of the endo-lysosomal “depots”. As a result, carrier free siRNA has reduced potency vs transfection agent delivery in short-term in vitro studies. Importantly, carrier free conjugates can be just as potent and often longer-lasting in vivo in the absence of an endosome disrupting agent. Please see: doi: [10.1093/nar/gkaa670](https://doi.org/10.1093/nar/gkaa670).

8. Could the increased accumulation of the divalent lipid-siRNA conjugate in the lungs (Supplementary Figure 3) reflect a degree of serum-induced aggregation of the more hydrophobic conjugate compared to a monovalent lipid conjugate?

We do not believe that is the case. If the hydrophobic conjugates were aggregating, we would be able to visualize that in the vasculature in our intravital microscopy measurements. We would

also expect to be able to capture such a phenomenon with our SEC experiments. We have now added the SEC traces comparing the monovalent and divalent conjugates in **Supplementary Fig. S3E** to demonstrate that the binding profiles are comparable, indicating that divalent conjugate aggregation does not account for the difference in biodistribution.

9. In the conclusion, the authors state “Overall, this work has important implications for the delivery of siRNAs to extrahepatic targets, a goal that has remained clinically elusive”. There are numerous reports in the literature of siRNA-based effects in extrahepatic tumour models. The authors should refine this statement.

We would like to emphasize the use of the word “clinically” in our statement. We intended for this statement to acknowledge that siRNA delivery to extrahepatic sites (e.g., tumors) has only achieved preclinical, rather than clinical, success. We aim to convey that our work herein adds to the body of knowledge that seeks to bridge this gap.

10. MCL-1 or MCL1 should be written consistently in the manuscript as write both
We have reviewed the text to ensure that when we are describing the gene, we have used *MCL1*, and when we are referring to the protein, we have used MCL-1.

Reviewer #2 - Breast cancer therapy (Remarks to the Author):

In this paper entitled “structural optimization of siRNA conjugates for albumin binding achieves effective MCL1-directed cancer therapy”, the author presented a systemic analysis of lipid library modified siRNA for enhanced binding to albumin as an efficient method for siRNA delivery platform. The author synthesized a series of various lipid combination as an optimization technique for obtaining the best siRNA-binding albumin platform. Although albumin-binding siRNA concept is not new, the author did bring to light the underlying mechanism of lipid-albumin binding mechanism that could be useful for future development of albumin-binding platform. Although the paper is logical and the author presented ample research results to prove their hypothesis, there are some crucial problems that need to be addressed prior to the consideration of acceptance of this paper.

Major comments:

1. The author strongly suggests that the albumin-binding of the lipid-modified siRNA could work better than siRNA-albumin direct link platform. If this is true, then, the lipid-linked siRNA (especially their optimized si<(EG18L)2) should show a preferentially higher albumin binding after systemic injection. The author should provide evidence that si<(EG18L)2 also works well after systemic injection (without prior albumin binding). This could be another significant result that could be further translated into clinical application.

As a point of clarification, si<(EG₁₈L)₂, and the other conjugates interrogated in this work are not pre-incubated with albumin prior to injection. Rather, they associate in situ with endogenous albumin. We apologize for the confusion and have clarified this in the introduction with the statement that we enabled “development of siRNAs that are intravenously injected carrier-free and can dock onto serum albumin as an endogenous carrier”.

2. The author mentioned that “the in vivo half-life of human albumin is an exceptional 19 days”. However, there is no indication that the author’s si<(EG18L)₂ system could be stable for siRNA (for 19 days) even after modification. Therefore, it is possible that in the systemic circulation, part of the si<(EG18L)₂ could be dissociated prior to the 19 days, with reduced efficacy. What is the half-life of si<(EG18L)₂ after dissociation from albumin in the circulation?

Based on the above comment, we believe that the reviewer is asking this question with the notion that our conjugate is premixed with exogenous albumin. However, this is not the case. We do show by SEC that the majority of the si<(EG₁₈L)₂ conjugate is albumin bound in the circulation after i.v. delivery. As an additional comment, there is a huge excess of albumin in the blood relative to the conjugate concentration, and we do anticipate that binding to endogenous albumin is a dynamic process wherein a single conjugate molecule can hop on and off of multiple albumin molecules during transit. It is not possible, ultimately, for us to decouple the circulation half-life of the bound vs. unbound structure. However, generally, non-conjugate siRNAs are rapidly filtered out through the kidneys in ~5 mins, and we anticipate the conjugate would suffer the same fate if not bound to any type of carrier protein / serum component.

3. Some of the comments made by the author are not supported by solid evidence. One of such examples is when the author claimed that “while biodistribution to other organ is similar, siRNA-L2 was modestly increased in lungs, a highly vascularized tissues, perhaps reflecting circulating conjugate at this time point”, and the author cited Figure S3). However, Fig S3 did not show the relationship between highly vascularized tissues with the amount of conjugate, whether or not it is conjugated. Please justify this claim.

We agree about the speculative nature of this comment and have removed it from the manuscript.

4. A similar problem is also found in the sentence of “... demonstrating progressively increasing t_{1/2} with increasing EG repeats.... Followed by ... suggesting a limit to the positive impact of increasing linker length on circulation half-life”. The author used EG18 followed by EG36 (or EG30, there are two different EGs given in Figure 2B-E, unsure if this is an error). There is no systemic analysis in between these two lengths (i.e. EG20, EG25, EG30), therefore, the statement is not entirely accurate.

We have amended this sentence to remove the suggestion that we have progressively and systematically increased the EG units. We now state more broadly that we observed a general increase in half-life with increased EG units.

5. In the sentence “to access the knockdown of the endogenous oncogenic driver MCL-1 in tumors...” the author mentioned that “...revealing dose-dependent MCL-1 knockdown at day 4 and day 8 (Fig. 3F). this sentence indicated two confusions: (i) the treatment was done twice at day 4 and day 8, or (ii) the knockdown only shows dose-dependency at day 4 and day 8, which were not true on both contexts. Please reword this sentence.

We have amended the text to read, “A single bolus was delivered intravenously to MDA-MB-231 tumor bearing mice (Fig. 3E), revealing dose-dependent MCL1 knockdown in tumors harvested at either 4 or 8 days after treatment (Fig. 3F), with as much as 85% MCL1 knockdown.”

6. In the subsection “albumin-binding increases carrier free tumor accumulation...”, the author is emphasizing on the point that the delivered si<(EG18L)2 is indeed dissociated from albumin in order for the siRNA to exert its effect. Therefore, exact measurement of dissociation would be crucial to prove this point, including whether or not the dissociation only occurs near/at the tumor?

We have modified this statement- the intention was to communicate a carrier in the traditional sense such as a nanoparticle. We understand that a reader could interpret the use of “carrier” here to refer to albumin. We have now changed the wording to “**Albumin-binding increases tumor accumulation of siRNA-lipid conjugates and tumor gene silencing.**” We would also like to emphasize that we did measure albumin association and dissociation constants by BLI and also quantify the relative amount of bound/unbound conjugate in plasma from i.v.-treated animals using SEC.

7. Although the binding (association) to albumin is important, the dissociation of siRNA from albumin is also important, in order for the siRNA to be uptaken into the cell cytoplasm. **At what condition will these si<(EG18L)2 dissociate from albumin? What is the underlying mechanism of the dissociation?** How can dissociated si<(EG18L)2 effectively enter cell cytoplasm without the help of targeting ligand or cell penetrating motif?

Albumin has natural fatty acid pockets that interact through non-covalent binding (Van der Vusse, *Drug metabolism and pharmacokinetics*, 24(4),300-307). To this end, there is a natural on rate and off rate involved in this type of binding. We demonstrated that dissociated si<(EG18L)2 enters the cell more effectively in the absence of albumin, though can also enter in albumin bound form (**Extended Data Figure 4**). The unbound conjugate can associate with cells even at 4 C, wherein endocytosis is inhibited. These results suggest that the lipid tails are able to bind to and potentially contribute to traversal of the cellular membrane to promote entry.

8. In figure 3F, please provide all concentrations for both treatment days. Current figure for Day 8 only has 10 and 20 mg/kg treatment data.

We only tested the two higher concentrations from day 4 to determine whether silencing was sustained at this dose level until day 8 for the sake of conserving animals.

9. In Figure 5, the in vivo analysis should be done in all three lipid variations. Current data only showed two lipid variation done on in vivo analysis. Furthermore, the si<(EG18Lunsat)2 albumin-bound ability being lesser than the other two is not fully convincing. The data presented have large variations inside the groups.

None of our experiments to survey the performance of this structure showed any indication that the unsaturated conjugate may be superior to the saturated one. We, therefore, determined that further analysis was outside of the scope of these studies.

Minor comments:

1. Please provide the country/region where the companies are based (in Materials and Methods).

We have added these details.

2. Please be careful with the labeling of figures. All figure should have their own labeling. For example, Figure 5 is missing two labels (one on the right of figure 5c and another one the right of figure 5e).

We have now added individual labels to the panel in Figure 5 and made this consistent in the manuscript.